# TESSERAQ: ULTRA LOW-BIT LLM POST-TRAINING QUANTIZATION WITH BLOCK RECONSTRUCTION

## ABSTRACT

Large language models (LLMs) have revolutionized natural language processing, albeit at the cost of immense memory and computation requirements. Post-training quantization (PTQ) is becoming the *de facto* method to reduce the memory footprint and improve the inference throughput of LLMs. In this work, we aim to push the upper limit of LLM PTQ by optimizing the weight rounding parameters with the block reconstruction technique, a predominant method in previous vision models. We propose TesseraQ, a new state-of-the-art PTQ technique, to quantize the weights of LLMs to ultra-low bits. To effectively optimize the rounding in LLMs and stabilize the reconstruction process, we introduce progressive adaptive rounding. This approach iteratively transits the soft rounding variables to hard variables during the reconstruction process. Additionally, we optimize the dequantization scale parameters to fully leverage the block reconstruction technique. We demonstrate that TesseraQ can be seamlessly integrated with existing scaling or clipping-based PTQ algorithms such as AWQ and OmniQuant, significantly enhancing their performance and establishing a new state-of-the-art. For instance, when compared to AWQ, TesseraQ improves the wikitext2 perplexity from 14.65 to 6.82 and average downstream accuracy from 50.52 to 59.27 with 2-bit weight-only quantization of LLaMA-2-7B. Across a range of quantization schemes, including W2A16, W3A16, W3A3, and W4A4, TesseraQ consistently exhibits superior performance.

## 1 INTRODUCTION

Large Language Models (LLMs) have revolutionized natural language processing with their remarkable capabilities. LLMs such as, GPT-4 (Bubeck et al., 2023) and LLaMA-3 (Meta, 2024), contain hundreds of billions of parameters. While this scale enables their impressive performance, it also poses significant deployment challenges (Zhou et al., 2024). LLMs require substantial memory and computational resources, making them impractical for many real-world applications, especially on consumer devices or in resource-limited environments (Dettmers et al., 2022). Quantization addresses this issue by reducing the precision of the model's parameters and activations, typically from 32-bit floating-point (FP32) to lower bit-width representations such as 8-bit or 4-bit integer (INT8, INT4). This technique significantly decreases the model's memory footprint to increase the I/O throughput, often with marginal performance loss.

Post-Training Quantization (PTQ) (Gholami et al., 2022) has perhaps become the most widespread and the easiest way to compress the LLM by reducing the bitwidth of the pretrained model's parameters. For example, with a single GPU and a small number of input sequences, GPTQ (Frantar et al., 2022) can compress an FP16 LLM into INT4 format by deriving the exact solution for quantization error minimization. Recent works like AWQ (Lin et al., 2023), QuaRot (Ashkboos et al., 2024) and OmniQuant (Shao et al., 2023) have pushed the compression limit further with INT3 weight-only quantization achieving a small performance gap with respect to the FP16 baseline. However, in a more challenging scenario like INT2 weight-only quantization and weight-and-activation quantization, these methods still incur a large performance gap compared to the original FP16 model.

We conjecture that the major reason for the low performance on ultra low-bit PTQ is limited optimization space. Most works only focus on optimizing distribution transformation or weight clipping ranges (Lin et al., 2023; Wei et al., 2023; Shao et al., 2023). While being straightforward, they prove inadequate for extremely low-bit scenarios due to the constrained optimization space. For instance,

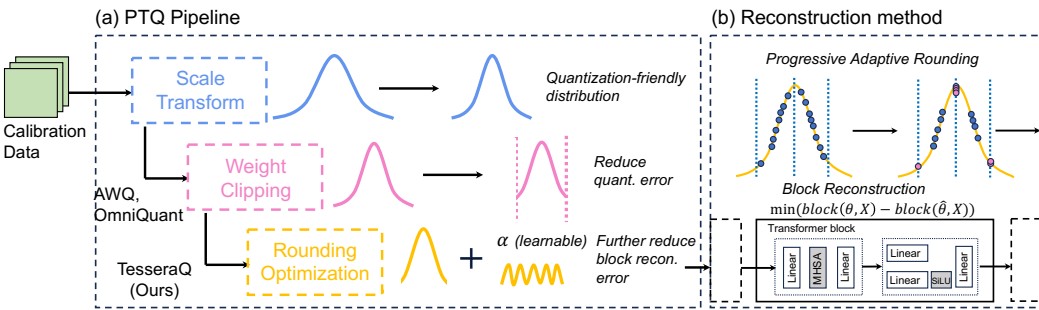

Figure 1: **The overall workflow of our proposed method**. (a) We apply TesseraQ to optimize the weight rounding parameters when the transformation scale and clipping range are determined using prior methods like AWQ/OmniQuant. (b) We propose Progressive Adaptive Rounding (PAR) for block-wise reconstruction, which iteratively hardens some rounding variables and optimizes the rest soft rounding variables till all variables become binary.

in per-channel weight quantization, a single clipping range or transformation scale must account for 4k∼20k weight elements in one channel, resulting in suboptimal quantization performance. We contend that to enhance LLM PTQ performance further, adjustment of the entire weight tensor is necessary. However, it is non-trivial to tune billions of parameters simultaneously.

To this end, we propose TesseraQ, a block reconstruction method tailored for LLM rounding optimization. We found that rounding optimization on a transformed and clipped LLM (Fig. 1(a)) brings significantly better performance than GPTQ. To accommodate the billions of parameter spaces in LLMs, our approach removes the dependency of regularization loss in the original rounding optimization processes (Nagel et al., 2020; Li et al., 2021) by introducing Progressive Adaptive Rounding (PAR). As shown in Fig. 1(b), PAR iteratively hard rounds certain rounding variables to binary values and optimizes the remainder to compensate for the rounding error. Moreover, we propose dequantization scale tuning to further decrease the reconstruction error. Leveraging block-wise reconstruction, we can efficiently and effectively optimize each LLM block on a single GPU. We have validated TesseraQ across various LLMs and uniform quantization bit-widths, demonstrating superior post-training performance and establishing new state-of-the-art quantized LLMs. We summarize our contributions as follows

1. We propose TesseraQ, a block reconstruction-based weight rounding optimization method for LLMs. TesseraQ can be combined with existing transformation or clipping methods like AWQ, OmniQuant, and QuaRot to obtain state-of-the-art results.

2. TesseraQ contains Progressive Adaptive Rounding and Dequantization Scale Tuning. Both methods can stabilize the reconstruction process and effectively optimize post-training performance.

3. Our method obtains state-of-the-art performance on both perplexity metric and zero-shot accuracy metric. For example, our method improves OmniQuant perplexity results from 37.4 to **8.0** on LLaMA-2-7B W2A16 quantization. Moreover, TesseraQ+QuaRot improves the average accuracy by **10%** on LLaMA-3.1-8B W3A3 quantization as compared to GPTQ+QuaRot.

## 2 PRELIMINARIES

This section briefly introduces the existing research directions in LLM PTQ. We adopt uniform affine quantization, which essentially discretizes the floating-point representation of weights/activations into low-bit fixed-point representation, given by

$$\mathbf{W}^q = \text{clamp}\left(\left\lfloor\frac{\mathbf{W}}{s}\right\rceil + z, 0, 2^N - 1\right), \text{ where } s = \frac{\gamma \max(\mathbf{W}) - \beta \min(\mathbf{W})}{2^N - 1}, z = -\left\lfloor\frac{\beta \min(\mathbf{W})}{s}\right\rceil.$$
(1)

where $s$ and $z$ denote the quantization step size and the zero point. The resulting $\mathbf{W}^q$ is in the INT-$N$ format. To restore it back to its original range, the dequantization step is given by $\hat{\mathbf{W}} = s \times (\mathbf{W}^q - z)$.

**Optimization Objective.** The plain rounding-to-nearest (RTN) method directly quantifies the model weights to integers without further optimization. However, this method usually results in

significantly low task performance. To improve the LLM PTQ performance, parameters related to quantization are optimized with different objectives. For example, GPTQ (Frantar et al., 2022) and AWQ (Lin et al., 2023) utilize the layer-wise reconstruction objective, given by

$$\min_{\epsilon}(\mathbf{L}(\theta + \epsilon) - \mathbf{L}(\theta)) \approx \sum_{\ell=1}^{L} \left|\left|\hat{\mathbf{W}}^{(\ell)}\mathbf{X}^{(\ell)} - \mathbf{W}^{(\ell)}\mathbf{X}^{(\ell)}\right|\right|_{F}^{2}, \tag{2}$$

where $\mathbf{L}$ is the loss function parameterized by weights in the whole model $\theta$ and quantization noise $\epsilon = \hat{\theta} - \theta$. $\ell \in \{1, 2, \ldots, L\}$ is the layer index and $\mathbf{X}$ is the input activations. While this layer-wise objective can provide efficient and exact solutions as in GPTQ, the objective does not consider inter-layer correlation like self-attention and residual connections in LLM. To this end, the block-wise reconstruction objective has been proposed (Li et al., 2021), as

$$\min_{\epsilon}(\mathbf{L}(\theta + \epsilon) - \mathbf{L}(\theta)) \approx \sum_{b=1}^{B} \left|\left|\mathrm{block}(\hat{\theta}^{(b)}, \mathbf{X}^{(b)}) - \mathrm{block}(\theta^{(b)}, \mathbf{X}^{(b)})\right|\right|_{F}^{2}. \tag{3}$$

where, block refers to one decoder block in LLMs comprising self-attention, projection, feed-forward and normalization layers. In practice, both layer-wise and block-wise objectives enable efficient calibration on a single GPU due to their local computation attributes. However, block-wise objectives exhibit better performance than layer-wise objectives as they better approximate the global loss (i.e., Eq. (3) left side) by accounting for contributions from multiple layers.

**Optimization Space.** Generally, three kinds of optimization spaces are explored in LLM PTQ, (1) the scale transformation, (2) the clipping range (i.e., finding the suitable $\gamma, \beta$), and (3) the weight values. They can be tied with either layer-wise or block-wise objectives. For instance, AWQ (Lin et al., 2023) and OS+ (Wei et al., 2023) optimize transformation and clipping range using Eq. (2), while OmniQuant (Shao et al., 2023) does similar optimization with Eq. (3). Since scale/clipping optimization methods are well-explored, in this paper, we aim to optimize weight values using block-wise objectives to further push the compression limits of LLM PTQ.

## 3 TESSERAQ: ULTRA LOW-BIT POST-TRAINING QUANTIZATION

### 3.1 PROBLEM STATEMENT

Element-wise weight adjustments were also studied in GPTQ (Frantar et al., 2022), in which the weights are computed using closed-form solutions using the inverse Hessian matrix. However, this technique makes it hard to improve scale-transformed models like AWQ[1]. Gong et al. (2024) also report similar observations with GPTQ. We hypothesize that the reason for the failed improvement of GPTQ+AWQ could be the layer-wise reconstruction objective and its approximation for the Hessian matrix, for example, to compute the inverse Hessian they dampen the matrix by $\lambda\mathbf{I}$.

In this work, we select a different weight optimization framework, the rounding optimization (Nagel et al., 2020; Li et al., 2021), which is a different optimization space compared to GPTQ, given by

$$\min_{\alpha} \left|\left|\mathrm{block}(\hat{\theta}, \mathbf{X}) - \mathrm{block}(\theta, \mathbf{X})\right|\right|_{F}^{2},$$
$$\text{s.t. } \hat{\theta} = \mathbf{s} \times (\theta^{q} - \mathbf{z}), \ \ \theta^{q} = \mathrm{clamp}\left(\lfloor\frac{\theta}{\mathbf{s}}\rfloor + \alpha + \mathbf{z}, 0, 2^{N} - 1\right), \ \ \alpha \in \{0, 1\}^{d}. \tag{4}$$

Here, $\theta$ denotes the total $d$ weight parameters of linear layers in the block, $\alpha$ is the rounding variable. Note that we omit the block index for simplicity. This rounding optimization framework shares both pros and cons. For its pros, rounding optimization space restricts the range of each weight parameter, allowing us to further improve the other PTQ models like AWQ/OmniQuant/QuaRot by slightly adjusting each weight element. As for its cons, there is no closed-form solution for rounding variables under the block-reconstruction framework. In addition, the binary rounding variables require either continuous relaxation with regularization loss (Nagel et al., 2020) or Straight-Through Estimator (Hubara et al., 2020) to be optimized. Thus, optimizing billions of rounding variables is challenging (Frantar et al., 2022). In Appendix A, we also show traditional rounding optimization is hard to scale on LLMs.

---

[1]GPTQ can be combined with rotation-transformed models like QuaRot. We also compare it in Sec. 4.

---

**Algorithm 1:** TesseraQ Calibration process

---

**Input:** FP16 LLM model; Calibration dataset, PAR iteration $K$, training steps $T$
**for** *all $b = 1, 2, \ldots, B$-th block in the FP model* **do**
    Collect input data to the block $\mathbf{X}$, the FP output $\mathrm{block}(\theta, \mathbf{X})$ ;
    Initialize rounding variable $\nu$, dequantization scale $\mathbf{v}$;
    **for** *all $k = 1, 2, \ldots, K$-iteration* **do**
        Calculate score (Eq. (6)) and hard-round the variables with lowest $P_k\%$ scores;
        **for** *all $t = 1, 2, \ldots, T$-training steps* **do**
            Gradient Descend Eq. (7) and update the soft rounding variables in this block as
            well as the dequantization scale;
    Set all rounding variables to 0/1 and merge them into original parameters;
**return** Quantized model;

---

### 3.2 PROGRESSIVE ADAPTIVE ROUNDING

To optimize $\alpha$, we introduce a differentiable rounding optimization framework called progressive adaptive rounding (PAR) that does not rely on regularization loss or the straight-through estimator in contrast to previous works (Nagel et al., 2020; Hubara et al., 2020).

To start with, we relax the rounding variable into a continuous variable by using the Sigmoid reparameterization $\alpha = \sigma(\nu)$. Therefore, $\nu$ can be initialized as $\nu = \sigma^{-1}(\theta/s - \lfloor \theta/s \rfloor)$, resulting in $\hat{\theta} = \theta$. The PAR algorithm divides all rounding variables into two sets: $\mathcal{S}_{\mathrm{Hard}}$ and $\mathcal{S}_{\mathrm{Soft}}$, standing for the *hard* and *soft* rounding of the variable $\nu$. Formally, we define the rounding function as

$$\alpha_i = \begin{cases} \sigma(\nu_i) = \frac{1}{1+\exp(-\nu)} & \text{if } i \in \mathcal{S}_{\mathrm{Soft}} \\ \sigma'(\nu_i) = \mathbf{1}_{\nu_i > 0} & \text{if } i \in \mathcal{S}_{\mathrm{Hard}} \end{cases}. \tag{5}$$

The $\sigma'(\nu_i)$ is a hard rounding function that returns 1 if $\nu_i$ is larger than 0, otherwise it returns 0. Starting from an empty hard rounding set, we iteratively put variables from $\mathcal{S}_{\mathrm{Soft}}$ into $\mathcal{S}_{\mathrm{Hard}}$ (called *Harden Phase*), and optimize the remaining soft rounding variables to compensate for the hard rounding loss (called *Soften Phase*). We elaborate on them in the following two subsections.

**Harden Phase.** Intuitively, after setting rounding variables to hard ones, we would expect minimum loss change in the block output error. Therefore, we define a score metric

$$HS(\nu) = |\sigma(\nu) - 0.5|. \tag{6}$$

Essentially, the lower the score, the closer the soft rounding variable ($\sigma(\nu)$) is to 0.5, implying that rounding these variables to binary values will result in a larger increase in reconstruction loss. As a result, in the Harden Phase, we sort the parameter indices based on their $HS$ and select the lowest $P\%$ of them to $\mathcal{S}_{\mathrm{Hard}}$. The hyper-parameter $P$ should increase from 0 to near 100 during block reconstruction. During the early stage of the reconstruction, $P$ can be increased rapidly, however, in the later stage, we slowly increase $P$ since the learnable soft variables are becoming fewer in each iteration. In our experiments, we find that TesseraQ is not sensitive to any specific decay schedule for $P$, as long as we progressively slow down the increasing rate of $P$. We conduct an ablation study of how to schedule the change of $P$ in Sec. 4.3.

**Soften Phase.** For this stage, we employ the gradient-descent optimization to optimize the soft rounding variable

$$\min_{\nu_i, i \in \mathcal{S}_{\mathrm{Soft}}} \left\| \mathrm{block}(\hat{\theta}, \mathbf{X}) - \mathrm{block}(\theta, \mathbf{X}) \right\|_F^2. \tag{7}$$

This objective can be optimized via gradient-based training like Adam (Kingma, 2014). During implementation, it would be too expensive to use masking to indicate soft rounding or hard rounding. Instead, for memory-efficient implementation, we can safely set the hard-rounding variables to $\infty$ or $-\infty$, which returns zero gradients in the sigmoid function. We find that optimizing Eq. (7) with nearly 200 steps can sufficiently decrease the block reconstruction error across different LLM models.

**Post-Processing.** After the entire PAR procedure is finished, we apply hard-rounding $\sigma'(\cdot)$ to all variables merge their values into the original weights, and then we can use the standard quantization formula (i.e., Eq. (1)). The merging can be effectively implemented by

$$\theta \leftarrow \theta + \mathbf{s} \times (\sigma'(\nu) - 0.5) \tag{8}$$

We provide a pseudocode for the learning process in Algorithm 1.

## 3.3 DEQUANTIZATION SCALE TUNING

During the PAR process, the quantized tensor $\theta^q$ undergoes continuous changes. To accommodate these dynamic adjustments, we propose a method that optimizes the dequantization scale concurrently with the rounding variable. Specifically, for the dequantization step, we introduce an additional parameter $\mathbf{v}$ and represent it as

$$\hat{\theta} = 2\sigma(\mathbf{v}) \times \mathbf{s} \times (\theta^q - \mathbf{z}). \tag{9}$$

By initializing $\mathbf{v} = \mathbf{0}$, we initialize the dequantization scale factor ($2\sigma(\mathbf{v})$) to 1 and subsequently adjust it to a value within the range $(0, 2)$. The sigmoid reparameterization can smooth the training process and reduce the efforts to adjust learning rate hyper-parameter. Note that we avoid optimizing the scale $s$ in the quantization step (Eq. (1)) since, (1) any change in $s$ would result in a change of the rounding mechanism (Nagel et al., 2020), (2) the optimization requires straight-through estimation (Shao et al., 2023) which leads to biased gradient calculation. Experiments in Sec. 4.3 demonstrate that dequantization scale tuning can benefit the final quantization performance of TesseraQ by a large margin.

## 4 EXPERIMENTS

### 4.1 EXPERIEMENTS SETUP

Most of our experiment setups are similar to OmniQuant (Shao et al., 2023), which also adopts block reconstruction loss function. Specifically, we employ asymmetric uniform quantization with 2/3/4-bit integers. We test both per-group and per-channel weight quantization. For example, we use the notation *W2A16g64* to denote the 2-bit per-group (group size is set to 64) weight-only quantization, where activations are FP16. In weight-activation quantization experiments (all INT precision), defaults are W4A4, W3A3, and W4A8 with per-channel weight and per-token activation quantization (Dettmers et al., 2022; Shao et al., 2023).

**Calibration Data and Comparison.** We report two types of evaluation metrics, the perplexity metric for evaluating the upstream datasets like WikiText2 (Merity et al., 2016), C4 (Raffel et al., 2020), and the average accuracy of 5 downstream reasoning tasks including PIQA (Bisk et al., 2020), ARC easy/challenge (Clark et al., 2018), WinoGrande (Sakaguchi et al., 2021) and HellaSwag (Zellers et al., 2019). The perplexity is evaluated with 2048 sequences. We use 512 2048-token segments from the WikiText2 training dataset as calibration data for perplexity comparison and for downstream task comparison, we sample same amount of calibration data from the C4 training dataset. We use `lm_eval` (ver0.4.2) to evaluate the accuracy.

**Training.** We set the total PAR number of iterations $K$ to 20 and gradually increase the $P_k$ from 0 to 100%. In each iteration, we optimize the learnable parameters ($\nu$ and $\mathbf{v}$) for 250 training steps. We use the Adam optimizer with a fixed learning rate of $1e-3$. We add $1e-4$ weight decay to $\mathbf{v}$ during training. The batch size is set to 4. We use AWQ transformation (Lin et al., 2023) to initialize our model since we find AWQ initialization is slightly better than OmniQuant across all configurations except W2A16 quantization. For W2A16, AWQ yields very high perplexity. Thus, in the W2A16 case, we directly use the pretrained OmniQuant model for initialization.

**Models and Baselines.** For the upstream tasks, we follow OmniQuant (Shao et al., 2023) to test weight-only quantization results on LLaMA-1-7B/13B/30B/65B (Touvron et al., 2023a), LLaMa-2-7B/13B/70B (Touvron et al., 2023b) and LLaMA-3-8B/70B (Meta, 2024). In this case, we compare GPTQ (Frantar et al., 2022), OmniQuant (Shao et al., 2023), AWQ (Lin et al., 2023), SignRound (Cheng et al., 2023) and GPTQ with QuaRot Ashkboos et al. (2024). For downstream tasks, we test LLaMA-2-7B, LLaMA-3.1-8B/70B across 5 downstream tasks. We compare GPTQ, AWQ, OmniQuant, and SignRound.

Table 1: **Weight-only quantization results of LLaMA-1/2/3 models**. We report WikiText2 perplexity (PPL ↓). *, †, ‡ means initialized from AWQ, OmniQuant, and QuaRot, respectively.

| LLaMA1&2 | Method | 1−7B | 1−13B | 1−30B | 1−65B | 2−7B | 2−13B | 2−70B | 3−8B | 3−70B |
|---|---|---|---|---|---|---|---|---|---|---|
| FP16 | - | 5.68 | 5.09 | 4.10 | 3.53 | 5.47 | 4.88 | 3.31 | 6.14 | 2.85 |
| W2A16 | GPTQ | 2.1e3 | 5.5e3 | 499.75 | 55.91 | 7.7e3 | 2.1e3 | 77.95 | 8.4e4 | 1.6e4 |
| | GPTQ‡ | 11.13 | 9.14 | 7.04 | 5.91 | 18.77 | 10.84 | 5.68 | 24.98 | 16.29 |
| | AWQ | 1.1e5 | 7002 | 1.2e5 | 6.3e6 | 2.9e6 | 6.2e3 | 3973 | 4.1e5 | 8.6e4 |
| | OmniQuant | 15.47 | 13.21 | 8.71 | 7.58 | 37.37 | 17.21 | 7.81 | - | - |
| | **TesseraQ†** | **7.56** | **6.56** | **5.75** | **5.21** | **8.05** | **6.55** | **5.26** | **17.88‡** | **11.56‡** |
| W2A16 g128 | GPTQ | 44.01 | 15.60 | 10.92 | 9.51 | 36.77 | 28.14 | NAN | 226.7 | 16.06 |
| | GPTQ‡ | 16.25 | 8.14 | 6.62 | 5.61 | 16.10 | 9.29 | 5.32 | 17.43 | 30.89 |
| | AWQ | 13.08 | 10.02 | 7.46 | 6.08 | 14.65 | 8.93 | 5.72 | 334.1 | 10.98 |
| | SignRound | 641.8 | 8.36 | 7.13 | 5.52 | NAN | 7.64 | NAN | - | - |
| | OmniQuant | 9.72 | 7.93 | 7.12 | 5.95 | 11.06 | 8.26 | 6.55 | - | - |
| | **TesseraQ*** | **6.92** | **6.07** | **5.26** | **4.83** | **6.82** | **5.92** | **4.73** | **10.03** | **7.47** |
| W2A16 g64 | GPTQ | 22.10 | 10.06 | 8.54 | 8.31 | 20.85 | 22.44 | NAN | 86.32 | 11.78 |
| | GPTQ‡ | 11.44 | 7.70 | 6.23 | 5.26 | 15.30 | 9.17 | 5.19 | 16.58 | 21.50 |
| | AWQ | 10.65 | 8.66 | 6.65 | 5.58 | 11.87 | 7.81 | 5.30 | 53.07 | 9.04 |
| | OmniQuant | 8.90 | 7.34 | 6.59 | 5.65 | 9.62 | 7.56 | 6.11 | - | - |
| | **TesseraQ*** | **6.78** | **5.97** | **5.18** | **4.70** | **6.67** | **5.81** | **4.60** | **9.28** | **6.96** |
| W3A16 | GPTQ | 8.06 | 6.76 | 5.84 | 5.06 | 8.37 | 6.44 | 4.82 | 16.84 | 18.94 |
| | GPTQ‡ | 6.15 | 5.45 | 4.53 | 4.01 | 6.13 | 5.35 | 3.72 | 7.54 | 5.22 |
| | AWQ | 8.49 | 6.38 | 5.89 | 6.03 | 14.17 | 6.42 | 4.22 | 11.79 | 12.28 |
| | OmniQuant | 6.49 | 5.68 | 4.74 | 4.04 | 6.58 | 5.58 | 3.92 | - | - |
| | **TesseraQ*** | **5.99** | **5.35** | **4.44** | **3.89** | **5.84** | **5.16** | **3.68** | **7.46** | **5.12** |
| W3A16 g128 | GPTQ | 6.55 | 5.62 | 4.80 | 4.17 | 6.29 | 5.42 | 3.85 | 9.58 | 5.25 |
| | GPTQ‡ | 6.07 | 5.41 | 4.48 | 3.92 | 5.99 | 5.28 | 3.65 | 7.42 | 4.98 |
| | AWQ | 6.38 | 5.52 | 4.59 | 3.92 | 6.19 | 5.30 | 3.72 | 8.24 | 4.63 |
| | SignRound | 6.28 | 5.45 | 4.50 | 3.90 | 8.09 | 5.23 | 3.68 | - | - |
| | OmniQuant | 6.15 | 5.44 | 4.56 | 3.94 | 6.03 | 5.28 | 3.78 | - | - |
| | **TesseraQ*** | **5.95** | **5.32** | **4.40** | **3.82** | **5.71** | **5.11** | **3.61** | **6.90** | **4.13** |
| W4A16 | GPTQ | 6.13 | 5.40 | 4.48 | 3.83 | 5.83 | 5.13 | 3.58 | 7.28 | 4.94 |
| | GPTQ‡ | **5.78** | 5.20 | 4.24 | 3.65 | 5.61 | 5.00 | 3.42 | 6.57 | 3.59 |
| | AWQ | 5.99 | 5.24 | 4.30 | 3.71 | 5.82 | 5.07 | 3.49 | 7.09 | 5.19 |
| | OmniQuant | 5.86 | 5.21 | 4.25 | 3.71 | 5.74 | 5.02 | 3.47 | - | - |
| | SignRound | 5.93 | 5.21 | 4.23 | 3.65 | 5.81 | 5.00 | **3.40** | - | - |
| | **TesseraQ*** | **5.78** | **5.17** | **4.20** | **3.63** | **5.56** | **4.96** | **3.40** | **6.48** | **3.33** |

## 4.2 MAIN RESULTS

**Perplexity Evaluation.** We summarized the Wikitext2 perplexity (PPL) results in Table 1. Our method consistently outperforms existing methods like AWQ and OmniQuant, particularly for the low-bit W2A16 configuration. Remarkably, in the W2A16 case, all existing methods except OmniQuant and GPTQ with QuaRot failed to successfully quantize the models (yielding $> 1e3$ perplexity). On the LLaMA-2-7B model, OmniQuant only obtains 37.37 PPL while our method largely improves this result to **8.05**. In addition, LLaMA-3-8B demonstrates extremely low quantization resiliency, where the AWQ model crashed in W2A16g128 quantization. Our method, on the other hand, significantly improves the wikitext2 PPL from 334 to 10.03. We observe that in general, the lower the bitwidth, the more improvement we can obtain from TesseraQ. This confirms our initial intuition that extremely low-bit weight quantization requires a thorough adjustment of each weight element. Additionally, the C4 (Raffel et al., 2020) PPL results are provided in Appendix: Table 9. Note that the C4 results for OmniQuant are re-evaluated from the official checkpoint to align the evaluation protocol. Overall, C4 PPL results concur with the Wikitext2 results, demonstrating a similar trend in performance improvement. For example, TesseraQ improves the PPL of LLaMA-2-7B model from 90.64 to 14.82 with W2A16 quantization.

**Downstream Tasks Evaluation.** We also test the weight-only quantization performance on five reasoning tasks. The results are summarized in Table 2, for LLaMA-2-7B, LLaMA-3.1-8B/70B[2]. Notably, we found that the LLaMA-3.1-8B model demonstrates low quantization resiliency, as also shown in Huang et al. (2024c). For example, with W2A16g128 AWQ, this model drops more than 30% average accuracy on downstream tasks, while the gap is 15% for the LLaMA-2-7B model. Fortunately, our TesseraQ can substantially increase the average performance on the downstream

---

[2]We did not implement OmniQaunt on LLaMA-3.1 models due to its high resource & time demands.

Table 2: **Weight-only quantization Results of various LLMs**. We report the accuracy of 5 common sense reasoning tasks ($\uparrow$). * means initialized from AWQ.

| Models | Bitwidths | Methods | PiQA | ArcE | ArcC | HellaSwag | WinoGrande | Avg. |
|---|---|---|---|---|---|---|---|---|
| LLaMA-2-7B | FP16 | - | 78.07 | 76.34 | 43.51 | 57.17 | 69.21 | 64.87 |
| | W2A16 g128 | GPTQ | 58.21 | 33.75 | 19.79 | 29.60 | 51.30 | 38.53 |
| | | AWQ | 67.73 | 55.47 | 28.74 | 41.37 | 59.27 | 50.52 |
| | | OmniQuant | 64.79 | 51.13 | 24.83 | 40.30 | 56.90 | 47.59 |
| | | SignRound | 72.96 | 65.99 | 32.25 | 47.35 | 61.01 | 55.92 |
| | | TesseraQ* | 75.13 | 70.03 | 35.83 | 50.17 | 65.19 | 59.27 |
| | W3A16 g128 | GPTQ | 76.65 | 73.69 | 40.52 | 54.43 | 66.61 | 52.39 |
| | | AWQ | 76.71 | 73.56 | 41.63 | 54.79 | 67.64 | 62.87 |
| | | OmniQuant | 76.93 | 74.66 | 39.59 | 54.95 | 67.16 | 62.66 |
| | | SignRound | 76.82 | 75.25 | 42.92 | 55.33 | 68.27 | 63.72 |
| | | TesseraQ* | 77.58 | 74.45 | 41.46 | 55.47 | 68.90 | 63.59 |
| LLaMA-3.1-8B | FP16 | - | 80.08 | 81.43 | 51.19 | 59.95 | 73.55 | 69.25 |
| | W2A16 g128 | GPTQ | 53.86 | 26.55 | 20.64 | 27.87 | 53.35 | 36.46 |
| | | AWQ | 57.34 | 35.18 | 18.26 | 28.05 | 53.27 | 38.42 |
| | | TesseraQ* | 75.68 | 68.98 | 35.66 | 50.21 | 66.29 | 59.37 |
| | W3A16 g128 | GPTQ | 77.53 | 75.04 | 43.60 | 56.15 | 71.66 | 64.80 |
| | | AWQ | 77.91 | 77.77 | 44.62 | 54.89 | 70.56 | 65.15 |
| | | TesseraQ* | 79.27 | 79.46 | 47.35 | 57.80 | 72.93 | 67.36 |
| LLaMA-3.1-70B | FP16 | - | 83.13 | 87.12 | 60.92 | 66.47 | 79.56 | 75.44 |
| | W2A16 g128 | GPTQ | 65.83 | 49.54 | 26.19 | 42.74 | 61.33 | 49.11 |
| | | AWQ | 73.45 | 68.01 | 40.27 | 48.11 | 62.19 | 58.40 |
| | | TesseraQ* | 78.23 | 78.70 | 47.35 | 57.91 | 71.74 | 66.79 |
| | W3A16 g128 | GPTQ | 80.79 | 82.70 | 55.54 | 63.18 | 77.03 | 71.85 |
| | | AWQ | 81.72 | 84.89 | 55.98 | 63.71 | 78.68 | 72.99 |
| | | TesseraQ* | 82.86 | 85.52 | 58.70 | 64.99 | 78.37 | 74.09 |

tasks, bringing the gap between W2 and FP16 to only 9%. TesseraQ also outperforms a recent rounding optimization method, SignRound (Cheng et al., 2023), for W2A16g128, demonstrating the effectiveness of our method.

**Weight-Activation Quantization Evaluation.** Finally, we test weight-activation quantization scenarios with per-channel weight quantization and per-token activation quantization. With quantized activations, the inference speed of LLMs on GPUs/TPUs can be improved especially in the prefill stage (Lin et al., 2023). We experiment with W4A4, and W4A8 quantization and compare with three baselines, SmoothQuant, OS+, AWQ, QLLM (Liu et al., 2023a). The results are provided in Table 3. Table 3 summarizes the perplexity on WikiTex2, C4 and average accuracy on downstream tasks. (The detailed accuracy of each downstream task is located in Appendix: Table 13.) We observe a consistent improvement of 7% accuracy with TesseraQ compared to AWQ. Additionally, we also combine our method with a recent rotation-based quantization method, QuaRot (Ashkboos et al., 2024), and compare QuaRot+GPTQ and QuaRot+TesseraQ with W4A4 and W3A3 quantization. Combined with QuaRot, TesseraQ also exceeds GPTQ by 10% accuracy on the 8B model with W3A3 quantization, demonstrating the superiority of TesseraQ.

**Results on Mistral-7B.** Additionally, we also test the performance of our method on the Mistral-7B model Jiang et al. (2023), which achieves high pretrained accuracy and demonstrates higher quantization resiliency. We test its weight-only quantization (W2A16g128, W3A16g128) and weight-activation quantization (W4A4, W4A8) performance in the Appendix (Table 12). Our TesseraQ consistently outperforms other methods like SignRound, AWQ, and GPTQ.

## 4.3 ABLATION STUDIES

Below ablation studies are conducted with the LLaMA-2-7B model with W2A16g128 quantization.

**Calibration Data.** In this section, we compare the performance of different calibration datasets and sizes. We sample calibration data from either WikiText2 (Merity et al., 2016) or C4 (Raffel et al., 2020) training dataset. We also experiment with the different sample sizes, ranging from 128 to 512. Meanwhile, we change the batch size during rounding optimization, ranging from 1 to 4.

Table 3: **W4A4/W3A3 quantization results of LLaMA-1/2/3**. We use per-channel weight quantization and per-token activation quantization *, [†] means initialized from AWQ, QuaRot.

| Bitwidths | Methods | LLaMA-7B | | | LLaMA-2-7B | | | LLaMA-3.1-8B | | |
|---|---|---|---|---|---|---|---|---|---|---|
| | | WT2($\downarrow$) | C4($\downarrow$) | Avg. ($\uparrow$) | WT2($\downarrow$) | C4($\downarrow$) | Avg. ($\uparrow$) | WT2($\downarrow$) | C4($\downarrow$) | Avg. ($\uparrow$) |
| FP16 | Pretrained | 5.68 | 7.08 | 62.30 | 5.47 | 6.97 | 64.87 | 6.24 | 9.54 | 69.25 |
| W4A4 | SmoothQuant | 585.1 | 780.5 | 37.02 | NAN | NAN | 35.29 | 654.6 | 508.5 | 36.09 |
| | OS+ | 16.47 | 25.51 | 50.50 | 29.03 | 39.71 | 50.50 | 124.2 | 67.44 | 40.71 |
| | AWQ | 12.67 | 17.58 | 48.41 | 13.87 | 19.54 | 49.99 | 60.99 | 74.08 | 42.51 |
| | OmniQuant | 11.26 | 14.51 | 50.47 | 14.61 | 18.39 | 49.86 | - | - | - |
| | QLLM | 9.65 | 12.29 | 51.84 | 11.75 | 13.26 | 51.60 | - | - | - |
| | TesseraQ* | **8.90** | **12.29** | **55.45** | **9.18** | **12.55** | **55.12** | **25.73** | **30.71** | **50.87** |
| | Atom (W4A4g128) | 6.16 | 7.70 | 60.17 | 6.14 | - | - | - | - | - |
| | QuaRot | 8.37 | 11.44 | 55.38 | 14.19 | 19.72 | 47.57 | 17.83 | 28.08 | 51.83 |
| | GPTQ[†] | **6.16** | 8.37 | 61.37 | **6.16** | 8.44 | 61.45 | 8.39 | 13.24 | 62.87 |
| | TesseraQ[†] | 6.27 | **8.07** | **61.92** | 6.23 | **8.23** | **61.75** | **8.05** | **12.62** | **65.12** |
| W3A3 | Atom (W3A3g128) | 11.77 | 15.43 | 49.28 | - | - | - | - | - | - |
| | QuaRot | 2315 | 1665 | 35.53 | 10996 | 10940 | 35.18 | 91551 | 65662 | 35.25 |
| | GPTQ[†] | 11.57 | 13.89 | 50.82 | 14.54 | 20.76 | 44.62 | 93.08 | 104.73 | 37.87 |
| | TesseraQ[†] | **10.79** | **13.68** | **51.10** | **13.90** | **15.08** | **50.13** | **27.80** | **30.81** | **47.33** |

Table 4: **Ablation studies of calibration data source and data sizes**. We report the LLaMA-2-7B W2A16g128 quantization results with task performances and calibration costs.

| #Samples | BS | Runtime/ | Calib. Data: WikiText2 | | | Calib. Data: C4 | | |
|---|---|---|---|---|---|---|---|---|
| | | GPU Mem. | WikiText2($\downarrow$) | C4($\downarrow$) | Avg.($\uparrow$) | WikiText2($\downarrow$) | C4($\downarrow$) | Avg.($\uparrow$) |
| 128 | 1 | 3.2h/17.5GB | 7.33 | 11.39 | 56.58 | 8.54 | 10.83 | 56.87 |
| 256 | 2 | 3.9h/28.6GB | 7.10 | 11.16 | 57.17 | 8.32 | 10.66 | 57.85 |
| 512 | 2 | 4.0h/40.4GB | 7.14 | 11.22 | 57.42 | 8.22 | 10.47 | 58.56 |
| 512 | 4 | 6.0h/65.4GB | **6.82** | 10.77 | 58.35 | 8.05 | **10.29** | **59.27** |

Table 4 demonstrates the task performance (PPL and average accuracy metric) as well as the calibration costs (algorithm runtime and GPU memory footprint). First, we find that the source of calibration data will impact the perplexity evaluation. The performance benefits if evaluation data and calibration data are from the same dataset. For example, the C4-calibrated model has 1.2 higher WikiText2 PPL than the WikiText2-calibrated model. Second, increasing the number of samples and the batch size consistently improves the task performance. However, it may also lead to higher runtime and GPU memory consumption, which may be alleviated via multi-GPU calibration. Nevertheless, it is worthwhile to note that even with 128 samples and a batch size of 1, our TesseraQ can significantly improve the baseline AWQ results.

**Algorithm choices.** We also test the algorithm choices in TesseraQ . To be more specific, we experiment with block reconstruction with or without progressive adaptive rounding (PAR) and dequantization scale tuning (DST) and compare their final task performance. As shown in Table 5, both PAR and DST contribute a lot to the final perplexity metric (denoted by WT2 (WikiText2) and C4) and average accuracy (denoted by Avg.). Remarkably, applying one of them solely can also improve the AWQ baseline (first row) results by a large margin.

Table 5: TesseraQ Algorithm choices.

| PAR | DST | WT2 | C4 | Avg. |
|---|---|---|---|---|
| ✗ | ✗ | 14.65 | 18.67 | 50.52 |
| ✓ | ✗ | 7.72 | 11.95 | 56.79 |
| ✗ | ✓ | 8.58 | 13.14 | 54.45 |
| ✓ | ✓ | **6.82** | **10.77** | **58.35** |

**PAR Schedule.** We investigate how to adjust the $P$ during progressive adaptive rounding. In our implementation, we use a handcrafted design, which manually decreases the soft rate (i.e., the percentage of soft rounding variable) as shown in Fig. 2. Our handcrafted design gradually decays the soft rate. To demonstrate that our PAR is quite robust to the schedule of soft rate, we also test several rule-based adjustments, which adjust the soft rate as $\frac{1}{\exp(tx)}$, where $x \in (0, 1]$ is the scaled iteration number and $t$ is the temperature hyper-parameter. We test $t = \{2, 3, 4, 5, 6, 7\}$ and compare it with our handcrafted implementation with LLaMA-2-7B W2A16g128 quantization. The results in Fig. 2 show that $t = 4, 5$ and our handcrafted adjustments obtain the best performance. Overall, we find that our algorithm is not sensitive to the scheduling, and has consistently superior performance than the AWQ initialized model.

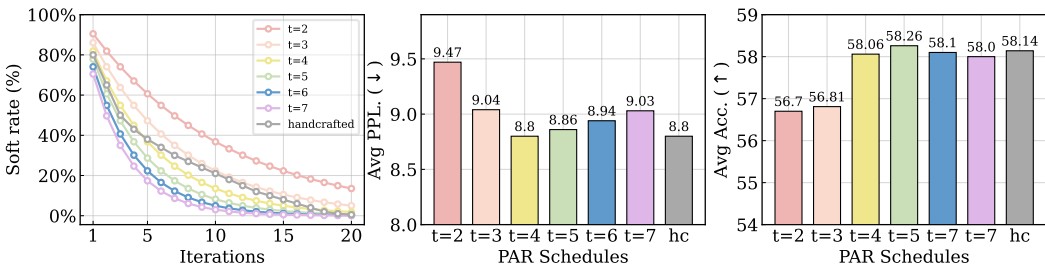

Figure 2: **Ablation study of PAR schedule.** We experiment several rule-based $P$ adjustments and one handcrafted adjustment. *(AWQ baseline results: average PPL: 16.66, average acc.: 50.52).*

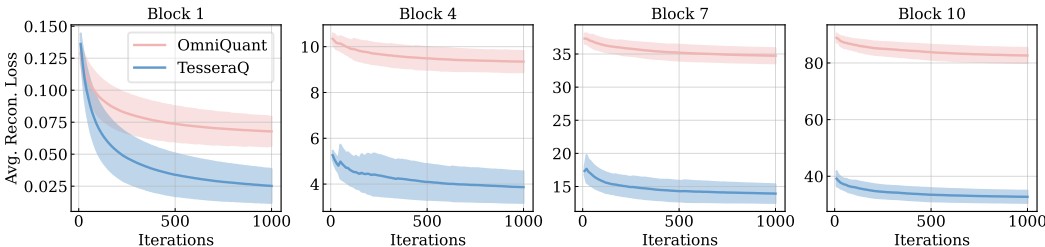

Figure 3: **Reconstruction loss convergence.** We compare the block reconstruction loss of Omni-Quant and TesseraQ during optimization. Our method significantly reduces the loss in each block.

## 4.4 VISUALIZATION

In this section, we provide visualizations of our calibration process to interpret the effectiveness of our method. The experiments are conducted on LLaMA-2-7B with W2A16g128 quantization. We first compare the loss convergence value in OmniQuant and TesseraQ, both of which calibrate the model with block reconstruction loss. To ensure a fair comparison, we use the same AWQ initialization to these two methods and align all training hyper-parameters. As shown in Fig. 3, during the first block reconstruction, TesseraQ reduces more loss than OmniQuant. In the following blocks, the loss gap between our method and Omniquant keeps on increasing. Consequently, TesseraQ will have a much lower model output error due to the cumulative effect of reconstruction.

Since rounding variables ($\alpha$) are binary, we also demonstrate the number or percentage of rounding variables that flip after TesseraQ. In Table 6, we show the number and the percentage of flipped variables. Overall we observe around 3%~8% of variables flip, amounting to over 10M parameters per block. This proves that tensor-wise adjustment can be used to significantly improve previous scale transformation adjustments like AWQ. We also found that attention layers tend to have less flipped rounding compared to MLP layers. 2/3-bit quantization also flips more than 4-bit quantization.

## 4.5 HARDWARE EVALUATION

To demonstrate the weight compression effect and the inference throughput change, we test LLaMA-3.1-8B/70B/405B under different GPU environments, kernel backend and different bitwidths. Table 7 summarizes the results of inference throughput (generated token per second) with batch size 1 or 16. Remarkably, W2A16g128 reduces the weight memory of the 405B model from 756GB to 114 GB and the 70B model from 132 GB to 21 GB. However, the INT2 dequantization kernel (in Triton (JonathanSalwan) support) is currently less optimized, especially for larger models, expending lower throughput compared to FP16. We find that INT4 with Exllama kernel can increase the throughput when batch size is 1 and achieve similar throughput with FP16 model when batch size is 16. Nonetheless, it is worthwhile to note that our TesseraQ complies with standard uniform quantization formats and can be deployed with various kernels that support uniform quantization on various devices, e.g., GPU, CPU, TPU, edge processor.

Table 6: **Number (percentages) of rounding variables that flip after TesseraQ**.

| Bits/Layers | q_proj | k_proj | v_proj | o_proj | gate_proj | up_proj | down_proj |
|---|---|---|---|---|---|---|---|
| W4A16g128 | 498k (2.97%) | 477k (2.85%) | 520k (3.10%) | 620k (3.70%) | 1.77M (3.92%) | 1.81M (4.02%) | 1.91M (4.24%) |
| W2A16g128 | 765k (4.55%) | 734k (4.37%) | 758k (4.52%) | 961k (5.73%) | 3.00M (6.67%) | 2.99M (6.64%) | 3.21M (7.12%) |

Table 7: **Comparison of weight memory compression and inference throughput**. We measure LLaMA-3.1 series model under various bitwidth/backend. WM stands for weight memory, $TP_n$ denotes inference throughput with a batch size of $n$ (output token/s).

| LLaMA-3.1 | | 8B (1×A5000) | | | 70B (2×A100-80GB) | | | 405B (4×A100-80GB) | | |
|---|---|---|---|---|---|---|---|---|---|---|
| BitWidth | Backend | WM | $TP_1$ | $TP_{16}$ | WM | $TP_1$ | $TP_{16}$ | WM | $TP_1$ | $TP_{16}$ |
| FP16 | Pytorch | 15GB | 49.23 | 358.1 | 132GB | 12.31 | 104.0 | 756GB | OOM | OOM |
| W4A16g128 | Exllama | 5.5GB | 57.54 | 361.1 | 39GB | 26.23 | 86.94 | 209GB | 7.01 | 18.59 |
| W2A16g128 | Triton | 3.9GB | 165.3 | 545.5 | 21GB | 4.93 | 54.35 | 114GB | 0.18 | 2.94 |

## 5 RELATED WORK

Quantization has been a primary method to compress and accelerate off-the-shelf large models. Survey papers by Gholami et al. (2022) and Nagel et al. (2021) have systematically summarized the progress of quantization. Here, we list several major quantization works, especially for LLMs.

**Post-Training Quantization for LLMs.** While Quantization aware Training (QAT) guarantees better task performance in low-bit quantization, PTQ is more suitable for LLM due to its less reliance on computing resources and training data. PTQ methods like Frantar et al. (2022); Lin et al. (2023); Wei et al. (2022; 2023); Shao et al. (2023); Chee et al. (2023); Liu et al. (2023a) improve the uniform quantization performance by optimizing weights, transformation scales, and clipping ranges. Our method continues improving the uniform quantization effect by incorporating rounding optimization. Other works try to improve PTQ in LLMs in different ways. For example, AQLM and GPTVQ (Egiazarian et al., 2024; van Baalen et al., 2024) explore non-uniform quantization schemes for weight-only quantization, which may better match the distribution of weights. LLM.int8 (Dettmers et al., 2022), BiLLM (Huang et al., 2024a), SiLLM (Huang et al., 2024b) apply mixed-precision quantization to keep salient weights in high precision and maintain the accuracy. However, these methods cannot be applied to quantize activations and thus cannot support integer MatMul. QuaRot (Ashkboos et al., 2024), SpinQuant (Liu et al., 2024) target activation outliers and eliminate them through the rotation matrix. We have demonstrated that our method can also be combined with them.

**QAT for LLM.** Recent works also explore QAT-based quantization for LLMs. To reduce data access, LLM-QAT (Liu et al., 2023b) generates language data for data-free QAT. To prevent massive weight memory usage, Q-LoRA (Dettmers et al., 2023) applies quantization-aware low-rank adaptation for finetuning. Recently, BitNet and BitNet b.158 (Wang et al., 2023; Ma et al., 2024) trained a 1-bit and 1.58-bit model from scratch, enabling multiplication-free LLM. However, these methods are hard to scale up due to the massive memory and computation requirements, especially for more than 70B models. As a result, they only focus on 1B∼3B-scale models.

## 6 CONCLUSION

In this paper, we have proposed TesseraQ, a PTQ method for effectively calibrating large language models. Based on block reconstruction, TesseraQ optimizes weight rounding through a progressive approach that iteratively hardens and softens the rounding variables. Together with dequantization scale tuning, TesseraQ can be seamlessly combined with other PTQ methods like transformation, clipping, and rotation, to reach new state-of-the-art performance. We demonstarte TesseraQ's superiority on open source LLaMA models. TesseraQ establishes a new state-of-the-art for quantized LLMs, in terms of perplexity, downstream accuracy and hardware performance.

**Limitations.** TesseraQ shares some limitations in terms of algorithm runtime, which may require longer processing time than existing baselines. For example, the LLaMA-2-7B takes 3∼6 hours to finish the calibration process, while for AWQ/GPTQ, the calibration time is around 0.5 hours. Nevertheless, compared to QAT, our method still exhibits remarkable resource efficiency in required data and GPU memory. We leave how to accelerate rounding optimization in our future directions.

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

## A   ABLATION STUDY ON ROUNDING OPTIMIZATION

In order to demonstrate the effectiveness of our proposed PAR, we compare our method with the several rounding optimization variants here.

**AdaRound** (Nagel et al., 2020). For AdaRound, the optimization is formulated by

$$\min_{\nu} \left\lVert \hat{\mathbf{W}}\mathbf{X} - \mathbf{W}\mathbf{X} \right\rVert_F^2 + \lambda \sum_{i,j} 1 - |2\sigma(\nu_{i,j}) - 1|^\beta,$$

$$\text{s.t. } \hat{\mathbf{W}} = \mathbf{s} \times (\mathbf{W}^q - \mathbf{z}), \quad \mathbf{W}^q = \text{clamp}\Big( \lfloor \frac{\mathbf{W}}{\mathbf{s}} \rfloor + \sigma(\nu) + \mathbf{z}, 0, 2^N - 1 \Big). \tag{10}$$

This method utilizes the layer-wise reconstruction objective and a regularization loss. Both $\lambda$ and $\beta$ control the strength of the regularization loss during optimization, which encourages the rounding variables to move towards 0 and 1.

**AdaQuant** (Hubara et al., 2020). This method directly utilizes the STE method to optimize the weighs, given by

$$\min_{\mathbf{V}} \left\lVert \hat{\mathbf{W}}\mathbf{X} - \mathbf{W}\mathbf{X} \right\rVert_F^2,$$

$$\text{s.t. } \hat{\mathbf{W}} = \mathbf{s} \times (\mathbf{W}^q - \mathbf{z}), \quad \mathbf{W}^q = \text{clamp}\Big( \lfloor \frac{\mathbf{W} + \mathbf{V}}{\mathbf{s}} \rceil + \mathbf{z}, 0, 2^N - 1 \Big), \frac{\partial \lfloor x \rceil}{\partial x} = 1. \tag{11}$$

Note that AdaRound and AdaQuant have not implemented their method on LLMs before. Therefore, we implement their method on our own and use the default hyper-parameters in their paper. Specifically, we experiment with the LLaMA-2-7B W2A16g128 quantization case, where the model is uniformly initialized from the AWQ checkpoint. Each weight tensor will be optimized for 5000 iterations for a fair comparison. We compare 3 methods, AdaRound, AdaQuant, and our PAR, with either layer-wise objective (Eq. (2)) or block-wise objective (Eq. (3)). For AdaRound, we set the learning rate the same as our method and while for AdaQuant the learning rate was 1e-5. The results are shown in the Table below.

Generally, we find that PAR consistently outperforms the other two rounding methods regardless of which objective. We think the reason is that we explicitly control the hardness of rounding variables through the progressive approach. While AdaRound and AdaQuant, they are less optimized on LLMs and may require more hyper-parameter search.

Table 8: **Ablation study on rounding method**.   The results are reported on LLaMA-2-7B W2A16g128 quantization.

| Rounding Method | Objective | WT2($\downarrow$) | C4($\downarrow$) |
|---|---|---|---|
| None (AWQ) | Layer | 14.65 | 18.67 |
| AdaRound | Layer | 10.68 | 15.67 |
| AdaQuant | Layer | 16.78 | 21.34 |
| PAR | Layer | 9.43 | 12.79 |
| None (OmniQuant) | Block | 11.06 | 16.34 |
| AdaRound | Block | 9.05 | 11.45 |
| AdaQuant | Block | 10.05 | 14.87 |
| PAR | Block | 6.82 | 10.77 |

## B   MORE EXPERIMENTAL RESULTS

In this section, we include additional experimental results from the main section.

### B.1   RESULTS ON C4

We demonstrate the perplexity results on the C4 datasets in Table 9. Note that the OmniQuant results are re-evaluated using the official checkpoint, which is slightly higher than the original paper results (Shao et al., 2023). Since the evaluation protocol can be different across different papers,

we ensure use of the same evaluation protocol to compare different methods. Note, we restrict all models here from using the WikiText2 calibration data as the calibration data will affect the perplexity metric as shown in our ablation study. The improvements of our method over existing approaches are consistent with the results on the WikiText2 dataset.

Table 9: **Weight-only quantization results of LLaMA-1 and LLaMA-2 Models**. We report C4 perplexity in this table. *, † means initialized from AWQ, and OmniQuant, respectively.

| LLaMA1&2 / PPL↓ | | 1−7B | 1−13B | 1−30B | 1−65B | 2−7B | 2−13B | 2−70B |
|---|---|---|---|---|---|---|---|---|
| FP16 | - | 7.08 | 6.61 | 5.98 | 5.62 | 6.97 | 6.46 | 5.52 |
| W2A16 | RTN | 1.3e5 | 5.6e4 | 2.7e4 | 2.2e4 | 4.8e4 | 7.2e4 | 2.4e4 |
| | GPTQ | 689.13 | 2.5e3 | 169.80 | 40.58 | NAN | 323.12 | 48.82 |
| | OmniQuant | 26.03 | 18.94 | 14.55 | 11.47 | 90.64 | 26.76 | 13.33 |
| | **TesseraQ†** | **13.28** | **11.43** | **10.81** | **8.52** | **14.82** | **11.96** | **9.15** |
| W2A16 g128 | RTN | 1.0e3 | 447.64 | 99.45 | 17.15 | 4.9e3 | 139.65 | 42.13 |
| | GPTQ | 27.71 | 15.29 | 11.93 | 11.99 | 33.70 | 20.97 | NAN |
| | AWQ | 16.35 | 12.93 | 10.07 | 8.78 | 18.67 | 11.88 | 8.49 |
| | OmniQuant | 14.06 | 11.27 | 10.37 | 8.65 | 16.34 | 12.14 | 9.33 |
| | **TesseraQ*** | **10.64** | **9.36** | **8.36** | **7.64** | **10.77** | **9.48** | **7.63** |
| W2A16 g64 | RTN | 151.43 | 76.00 | 30.07 | 11.34 | 475.35 | 28.69 | 13.43 |
| | GPTQ | 17.71 | 11.70 | 9.92 | 10.07 | 19.40 | 12.48 | NAN |
| | AWQ | 13.47 | 11.35 | 9.12 | 8.11 | 15.13 | 10.85 | 7.77 |
| | OmniQuant | 12.79 | 10.60 | 9.46 | 8.18 | 13.79 | 11.02 | 8.61 |
| | **TesseraQ*** | **10.32** | **9.05** | **8.18** | **7.48** | **10.50** | **9.23** | **7.44** |
| W3A16 | RTN | 28.26 | 13.22 | 28.66 | 12.79 | 402.35 | 12.51 | 10.02 |
| | GPTQ | 9.49 | 8.16 | 7.29 | 6.71 | 9.81 | 8.02 | 6.57 |
| | AWQ | 11.16 | 8.37 | 7.91 | 8.62 | 16.25 | 8.90 | 6.50 |
| | OmniQuant | 8.73 | 7.68 | 6.86 | 6.31 | 9.24 | 7.89 | 6.31 |
| | **TesseraQ*** | **8.15** | **7.38** | **6.60** | **6.16** | **8.30** | **7.41** | **6.08** |

**Results on Smaller-Size LLM for Edge Inference.** In addition to LLMs that are deployed on GPUs, we also test the performance of smaller-size LLMs geared for edge devices. We test LLaMA-3.2-1/3B models and compare them with AWQ in Table 10. We observe that our method significantly outperforms AWQ across different bitwidths in WikiText2 perplexity and average downstream task performance.

## B.2 W4A8 QUANTIZATION

We also provide the W4A8 quantization in Table 11. Overall we find a small difference in W4A8 quantization due to the 8-bit per-token activation quantization.

## B.3 DETAILED ACCURACY OF W4A4/W3A3 QUANTIZATION

Table 13 provides the detailed accuracy of each zero-shot tasks in W4A4/W3A3 quantization.

## B.4 EVALUATION ON GENERALIZATION

To validate the generalization capability of our method, we test the quantization on LLaMA-2chat (Touvron et al., 2023b), an instruction-tuned model for chatbots. Following OmniQuant experiments (Shao et al., 2023), we use GPT-4 evaluation protocol (Chiang et al., 2023), performance is assessed on the Vicuna benchmark, which comprises 80 questions. We compare our TesseraQ and OmniQuant on LLaMA-2-7B-chat with W3A16g128 quantization. Our model has a 69% win rate against the OmniQuant model. We also demonstrate some chat cases in Fig. 4.

Table 10: **Quantization Results of LLaMA-3.2 for edge inference**.

| Bitwidths | Methods | LLaMA-3.2-1B | | LLaMA-3.2-3B | |
|---|---|---|---|---|---|
| | | WT2($\downarrow$) | Avg. ($\uparrow$) | WT2($\downarrow$) | Avg. ($\uparrow$) |
| FP16 | Pretrained | 9.75 | 56.50 | 7.81 | 63.57 |
| W2A16g128 | AWQ | 5475 | 35.42 | 495.2 | 38.15 |
| | **TesseraQ*** | **18.61** | **43.36** | **11.94** | **51.53** |
| W3A16g128 | AWQ | 16.69 | 49.85 | 10.21 | 59.94 |
| | **TesseraQ*** | **11.08** | **53.24** | **8.45** | **61.58** |
| W4A16g128 | AWQ | 10.85 | 54.68 | 8.25 | 62.83 |
| | **TesseraQ*** | **10.09** | **54.98** | **7.96** | **63.63** |

Table 11: **Weight-activation quantization Results of various LLMs**. We report the accuracy of 5 reasoning tasks ($\uparrow$).

| Models | Bitwidths | Methods | PiQA | ArcE | ArcC | HellaSwag | WinoGrande | Avg. |
|---|---|---|---|---|---|---|---|---|
| LLaMA-7B | FP16 | - | 77.47 | 52.48 | 41.46 | 73.00 | 67.07 | 62.30 |
| | W4A8 | SmoothQuant | 75.19 | 70.45 | 37.45 | 51.06 | 64.87 | 59.81 |
| | | OS+ | 78.42 | 74.49 | 40.61 | 55.53 | 69.37 | 63.75 |
| | | AWQ | 77.63 | 73.31 | 41.89 | 55.50 | 69.85 | 63.65 |
| | | **TesseraQ*** | **78.89** | **75.33** | **41.55** | **56.11** | **69.14** | **64.21** |
| LLaMA-2-7B | FP16 | - | 78.07 | 76.34 | 43.51 | 57.17 | 69.21 | 64.87 |
| | W4A8 | SmoothQuant | 75.24 | 70.95 | 38.39 | 51.30 | 63.85 | 59.95 |
| | | Outlier Supp.+ | 77.09 | 74.74 | 42.57 | 56.37 | 68.51 | 63.86 |
| | | AWQ | 77.09 | 74.36 | 42.32 | 56.25 | 69.53 | 63.91 |
| | | **TesseraQ*** | **77.42** | **76.26** | **41.63** | **56.42** | **69.22** | **64.19** |
| LLaMA-3.1-8B | FP16 | - | 79.54 | 80.09 | 50.17 | 60.13 | 73.24 | 68.64 |
| | W4A8 | SmoothQuant | 71.98 | 66.37 | 34.55 | 50.46 | 67.40 | 58.16 |
| | | Outlier Supp.+ | 77.91 | 78.78 | 48.03 | 58.83 | 72.53 | 67.22 |
| | | AWQ | 79.00 | 78.40 | 48.63 | 58.81 | 72.45 | 67.46 |
| | | **TesseraQ*** | **78.99** | **79.88** | **47.61** | **59.09** | **72.77** | **67.67** |
| Mistral-8B | W4A8 | SmoothQuant | 79.59 | 77.56 | 46.50 | 57.62 | 71.11 | 66.48 |
| | | OS+ | 80.35 | 79.04 | 48.03 | 60.18 | 72.45 | 68.02 |
| | | AWQ | 79.92 | 79.79 | 47.35 | 58.80 | 74.26 | 68.03 |
| | | **TesseraQ*** | **80.36** | **79.92** | **49.57** | **60.54** | **73.79** | **68.84** |

Table 12: **Weight-activation quantization Results of Mistral-7B**. We report the accuracy of 5 reasoning tasks ($\uparrow$).

| Models | Bitwidths | Methods | PiQA | ArcE | ArcC | HellaSwag | WinoGrande | Avg. |
|---|---|---|---|---|---|---|---|---|
| Mistral-7B | FP16 | - | 80.68 | 80.93 | 50.42 | 61.26 | 73.79 | 69.42 |
| | W2A16 g128 | GPTQ | 64.20 | 45.74 | 22.35 | 36.68 | 55.02 | 44.80 |
| | | AWQ | 68.44 | 56.73 | 27.44 | 40.60 | 56.03 | 49.06 |
| | | SignRound | 75.84 | 70.88 | 30.73 | 50.87 | 62.90 | 58.24 |
| | | **TesseraQ*** | **76.87** | **71.67** | **39.59** | **54.09** | **68.11** | **62.07** |
| | W3A16 g128 | GPTQ | 79.70 | 78.70 | 48.41 | 59.15 | 71.98 | 67.19 |
| | | AWQ | 80.19 | 78.62 | 45.56 | 58.28 | 71.58 | 66.85 |
| | | SignRound | 79.54 | 78.70 | 46.33 | 59.60 | 72.85 | 67.40 |
| | | **TesseraQ*** | **79.59** | **78.36** | **47.44** | **59.87** | **71.98** | **67.45** |
| | W4A4 | SmoothQuant | 57.94 | 35.14 | 21.75 | 30.51 | 48.30 | 38.73 |
| | | OS+ | 66.70 | 56.73 | 30.20 | 42.39 | 52.01 | 49.61 |
| | | AWQ | 66.26 | 54.16 | 30.80 | 43.45 | 53.67 | 49.67 |
| | | **TesseraQ*** | **72.19** | **65.90** | **33.78** | **49.02** | **57.61** | **55.71** |

Table 13: **Detailed W4A4/W3A3 quantization results on each commonsense tasks of LLaMA-1/2/3**. We use per-channel weight quantization and per-token activation quantization *, † means initialized from AWQ, QuaRot.

| Models | Bitwidths | Methods | PiQA | ArcE | ArcC | HellaSwag | WinoGrande | Avg. |
|---|---|---|---|---|---|---|---|---|
| | FP16 | - | 78.67 | 75.33 | 41.80 | 56.96 | 69.85 | 64.53 |
| LLaMA-7B | W4A4 | SmoothQuant | 55.49 | 31.22 | 21.16 | 27.31 | 49.88 | 37.02 |
| | | OS+ | 67.46 | 57.74 | 31.05 | 41.83 | 54.38 | 50.50 |
| | | AWQ | 65.56 | 57.36 | 26.10 | 9.02 | 53.98 | 48.41 |
| | | OmniQuant | 66.15 | 45.20 | 31.14 | 56.44 | 53.43 | 50.47 |
| | | QLLM | 68.77 | 45.20 | 31.14 | 57.43 | 56.67 | 51.84 |
| | | **TesseraQ*** | **71.98** | **64.77** | **32.67** | **47.59** | **60.22** | **55.45** |
| | | Atom (W4A4g128) | 76.28 | 52.10 | 38.99 | 69.81 | 63.69 | 60.17 |
| | | QuaRot | 71.70 | 64.81 | 30.88 | 48.25 | 61.24 | 55.38 |
| | | GPTQ† | **76.55** | 72.60 | 37.11 | 53.67 | **66.93** | 61.37 |
| | | **TesseraQ†** | 76.22 | **73.31** | **39.25** | **54.45** | 66.38 | **61.92** |
| | W3A3 | Atom (W3A3g128) | 65.56 | 41.41 | 30.72 | 53.19 | 55.56 | 49.28 |
| | | QuaRot | 52.72 | 26.09 | 20.82 | 26.06 | 51.93 | 35.53 |
| | | GPTQ† | **68.98** | **58.92** | 26.87 | **43.90** | 55.40 | 50.82 |
| | | **TesseraQ‡** | 68.93 | 57.78 | **27.30** | 43.24 | **58.24** | **51.10** |
| LLaMA-2-7B | FP16 | - | 78.07 | 76.34 | 43.51 | 57.17 | 69.21 | 64.87 |
| | W4A4 | SmoothQuant | 53.04 | 25.71 | 20.22 | 25.71 | 51.77 | 35.29 |
| | | OS+ | 66.86 | 56.52 | 29.60 | 41.93 | 56.19 | 50.23 |
| | | AWQ | 64.80 | 53.87 | 30.20 | 43.11 | 57.93 | 49.99 |
| | | OmniQuant | 65.94 | 43.94 | 30.80 | 53.53 | 55.09 | 49.86 |
| | | QLLM | 67.68 | 44.40 | 30.89 | 58.45 | 56.59 | 51.60 |
| | | **TesseraQ*** | **70.89** | **63.34** | **32.93** | **48.28** | **60.14** | **55.12** |
| | | QuaRot | 66.54 | 55.51 | 25.76 | 37.80 | 52.25 | 47.57 |
| | | GPTQ† | **75.89** | 71.96 | 39.85 | **54.12** | 65.43 | 61.45 |
| | | **TesseraQ*** | 76.22 | **74.20** | 39.50 | 53.80 | 65.03 | **61.75** |
| | W3A3 | QuaRot | 51.74 | 25.54 | 22.86 | 25.84 | 49.88 | 35.18 |
| | | GPTQ† | 64.31 | 47.21 | 22.18 | 36.08 | 53.27 | 44.62 |
| | | **TesseraQ‡** | **68.28** | **56.82** | **28.58** | **41.96** | **55.01** | **50.13** |
| LLaMA-3.1-8B | FP16 | - | 79.54 | 80.09 | 50.17 | 60.13 | 73.24 | 68.64 |
| | W4A4 | SmoothQuant | 54.24 | 27.90 | 19.79 | 26.87 | 51.61 | 36.09 |
| | | OS+ | 57.34 | 40.99 | 20.22 | 33.19 | 51.77 | 40.71 |
| | | AWQ | 59.68 | 44.90 | 22.09 | 34.53 | 51.30 | 42.51 |
| | | **TesseraQ*** | 67.08 | 59.09 | 27.13 | 43.88 | 57.14 | 50.87 |
| | | QuaRot | 69.85 | 58.03 | 28.07 | 43.37 | 59.82 | 51.83 |
| | | GPTQ† | 76.22 | 73.94 | 41.21 | 55.47 | 67.48 | 62.87 |
| | | **TesseraQ†** | **77.64** | **77.27** | **44.80** | **56.03** | **69.85** | **65.12** |
| | W3A3 | QuaRot | 52.28 | 26.59 | 20.56 | 26.11 | 50.67 | 35.25 |
| | | GPTQ† | 56.96 | 33.62 | 20.47 | 28.87 | 49.40 | 37.87 |
| | | **TesseraQ†** | **66.05** | **51.59** | **24.40** | **40.59** | **53.98** | **47.33** |
| LLaMA-3.1-70B | FP16 | - | 83.13 | 87.12 | 60.92 | 66.47 | 79.56 | 75.44 |
| | W4A4 | SmoothQuant | 57.45 | 38.46 | 24.23 | 30.22 | 54.93 | 41.06 |
| | | OS+ | 53.04 | 25.79 | 22.01 | 25.88 | 48.85 | 35.12 |
| | | AWQ | 69.91 | 61.71 | 34.04 | 47.98 | 54.61 | 53.65 |
| | | **TesseraQ*** | 78.29 | 69.15 | 38.12 | 53.74 | 61.16 | 60.09 |
| | | QuaRot | 57.88 | 36.36 | 19.02 | 28.13 | 53.19 | 38.92 |
| | | GPTQ† | 79.76 | 80.17 | 50.59 | 60.71 | 73.08 | 68.87 |
| | | **TesseraQ†** | **81.84** | **82.64** | **54.07** | **63.90** | 65.64 | **69.62** |
| | W3A3 | QuaRot | 52.06 | 24.87 | 20.05 | 25.55 | 49.25 | 34.26 |
| | | GPTQ† | 55.98 | 34.80 | 19.45 | 28.38 | 51.46 | 38.02 |
| | | **TesseraQ†** | **74.80** | **66.03** | **36.42** | **51.34** | **58.43** | **57.42** |

**Question: What if the Internet had been invented during the Renaissance period?**

TesseraQ:

Ah, an intriguing question! If the Internet had been invented during the Renaissance period, it would have been a revolutionary technology that would have transformed the way people communicated, shared knowledge, and accessed information. Here are some ways it could have impacted society:

1. Faster dissemination of knowledge: The Renaissance was a time of immense cultural and intellectual growth, and the Internet would have accelerated the spread of new ideas and discoveries. Scholars and scientists could have shared their findings more quickly and easily, leading to a more rapid pace of innovation.

2. More accessible education: The Renaissance saw a renewed interest in classical learning, and the Internet would have made educational resources more widely available. People could have accessed educational content from all over Europe, without the need for expensive and time-consuming travel.

3. Enhanced cultural exchange: The Renaissance was a time of cultural exchange between Europe and other parts of the world, such as the Middle East and Asia. The Internet would have facilitated even more direct communication and collaboration between cultures, fostering greater understanding and mutual respect.

4. Improved communication: The Renaissance saw the development of new forms of communication, such as printing and the use of the Latin language. The Internet would have expanded upon these forms of communication, allowing people to share ideas and information more quickly and efficiently.

5. New forms of art and literature: The Renaissance saw a flowering of artistic and literary creativity, and the Internet would have provided new mediums for expression. People could have created digital art, music, and literature, and shared them with a global audience.

6. Increased access to information: The Renaissance saw a growing interest in the works of ancient Greece and Rome, and the Internet would have made it easier for people to access and study these texts.

7. More efficient commerce: The Renaissance saw the emergence of new trade routes and commercial practices, and the Internet would have further transformed the way people conducted business. Merchants could have easily accessed markets and customers all over Europe, and traded goods more efficiently.

8. New forms of entertainment: The Renaissance saw the emergence of new forms of entertainment, such as theater and music. The Internet would have provided new ways for people to enjoy and engage with these forms of entertainment, such as streaming and downloading music and videos.

Of course, it's important to note that the Internet as we know it today is a product of the Information Age, and its development was shaped by many factors beyond the Renaissance. However, it's fascinating to imagine how the Internet could have evolved if it had been invented during this time of cultural and intellectual flourishing.

OmniQuant:

Ah, an intriguing question! If the Internet had been invented during the Renaissance period, it would have likely been a vastly different technological landscape than what we have today. Here are a few ways the Internet might have evolved:

1. Renaissance Network: The Renaissance was a time of great cultural and intellectual exchange, and the Internet might have been conceived as a means of facilitating this exchange. It could have been a network of scholars and artists sharing ideas and creations through a series of interconnected nodes.

2. Print Press Reimagined: During the Renaissance, the printing press revolutionized the dissemination of information. An Internet-like network might have built upon this technology, with books, pamphlets, and other printed materials being distributed more widely and efficiently.

3. Urban Communication Networks: Cities like Florence, Venice, and Rome were hubs of trade and commerce. An Internet-like network might have emerged as a means of facilitating communication between these cities and their surrounding regions.

4. Artistic Collaboration: The Renaissance saw a renewed interest in classical art and culture. An Internet-like network might have fostered collaboration and exchange among artists, musicians, and writers, allowing them to share their work and ideas more widely.

5. Humanist Learning Network: The Renaissance was characterized by a renewed interest in classical knowledge and the pursuit of humanism. An Internet-like network might have facilitated the sharing of knowledge and ideas among scholars and thinkers, allowing them to build upon the achievements of previous generations.

6. Merchant Networks: The Renaissance saw the rise of trade and commerce, with merchants and traders traveling throughout Europe and beyond. An Internet-like network might have emerged as a means of facilitating communication and exchange between these merchants and traders.

7. Scientific Knowledge Sharing: The Renaissance saw a number of scientific breakthroughs and discoveries. An Internet-like network might have allowed scientists to share their findings more widely and rapidly.

8. Renaissance Revival: In the Renaissance, there was a renewed interest in classical culture and learning. An Internet-like network might have fostered a revival of this culture, allowing people to connect with and learn from the achievements of the past.

Of course, these are just a few possibilities, and the actual evolution of the Internet would likely have been shaped by a wide range of factors. However, it is fascinating to consider what might have been if the Internet had been invented during the Renaissance.

**Question: What if Christopher Columbus had not discovered the Americas?**

TesseraQ:

If Christopher Columbus had not discovered the Americas, the course of history would have been significantly different. Here are some possible outcomes:

1. Indigenous Cultures Continue Unimpeded: Without Columbus's arrival, the indigenous peoples of the Americas would have continued to thrive and develop their own cultures, traditions, and societies.

2. No European Settlements: Columbus's arrival led to the establishment of European settlements in the Americas, which had a profound impact on the indigenous populations. Without these settlements, the native cultures might have continued to flourish without interference from European colonizers.

3. Different Global Power Dynamics: The discovery of the Americas by Columbus marked the beginning of a new era of global power dynamics. Without this event, the balance of power between European nations and the indigenous peoples might have remained more stable, with fewer opportunities for European colonization and exploitation.

4. No Transatlantic Slave Trade: The transatlantic slave trade was fueled in part by the discovery of the Americas, as European powers sought to exploit the vast resources of the New World. Without Columbus's discovery, the transatlantic slave trade might have been less profitable, or might not have occurred at all.

5. Different Economic and Technological Developments: Columbus's arrival in the Americas led to the introduction of new technologies, crops, and economic systems. Without these introductions, the indigenous peoples might have continued to develop their own unique economic and technological systems, unimpeded by European influence.

6. No European Impact on Indigenous Languages: Columbus's arrival also led to the introduction of European languages, such as Spanish and English, which had a profound impact on the indigenous languages of the Americas. Without this influence, the indigenous languages might have continued to evolve and develop in their own unique ways.

7. Different Environmental Impacts: Columbus's arrival in the Americas also had significant environmental impacts, including the introduction of new diseases and the exploitation of natural resources. Without these impacts, the native ecosystems might have continued to thrive, with fewer disruptions from human activity.

Overall, it is impossible to say exactly how the world would have looked without Christopher Columbus's discovery of the Americas, but it is clear that the course of history was significantly altered by this event.

OmniQuant:

If Christopher Columbus had not discovered the Americas, it is likely that the indigenous peoples of the Americas would have continued to inhabit the lands without European colonization. The course of history would have been very different, with potentially significant impacts on the cultures, societies, and economies of both the indigenous peoples and the European powers.

Here are a few possible scenarios:

1. Continued Indigenous Sovereignty: Without Columbus's discovery, the indigenous peoples of the Americas may have continued to inhabit and govern their lands without European interference. This could have resulted in the preservation of their cultures, languages, and traditions.

2. Indigenous-European Interactions: Without the arrival of European colonizers, the indigenous peoples may have had more time to interact with European powers through trade or diplomacy. This could have led to a more peaceful coexistence between the two groups.

3. Different Technological Advancements: Without European contact, the indigenous peoples may have continued to develop their own technologies and societies at their own pace. This could have resulted in a more self-sufficient and sustainable way of life.

4. Alternative Global Power Dynamics: Without European colonization, the global power dynamics may have been very different. The indigenous peoples may have been able to establish their own empires or federations, or they may have been able to maintain their independence.

5. Unknown Impact on Ecosystems: Without European colonization, the ecosystems of the Americas may have been less impacted by invasive species, deforestation, and climate change. This could have resulted in a healthier and more resilient environment.

Overall, the consequences of Christopher Columbus's discovery of the Americas are complex and far-reaching, and it is difficult to predict exactly how things would have turned out without it. However, it is clear that the arrival of European colonizers had a profound impact on the indigenous peoples of the Americas, and their legacy continues to be felt today.

Figure 4: **Examples of chat cases on Vicuna Benchmark (Chiang et al., 2023).** We compare TesseraQ W3A16g128 and OmniQuan W3A16g128 quantization on LLaMA-2-7B-chat.

