# OpenReview forum: "TesseraQ: Ultra Low-Bit LLM Post-Training Quantization with Block Reconstruction"
_ICLR.cc/2025/Conference — Submitted to ICLR 2025_

### Official Review · Reviewer_wxS8 · 2024-10-16

**Soundness:** 2
**Presentation:** 3
**Contribution:** 2
**Rating:** 3
**Confidence:** 5

**Summary:**

TesseraQ is a new PTQ method for ultra-low bit quantization of LLM weights. It features progressive adaptive rounding and optimized dequantization scale parameters, improving block reconstruction. Integrated with PTQ algorithms like AWQ and OmniQuant, TesseraQ significantly enhances performance.

**Strengths:**

The authors introduce two techniques to optimize more parameters under PTQ settings, enhancing the performance of quantized models through block-wise reconstruction. These approaches, combined with existing transformation-based methods, deliver improved model accuracy without adding inference overhead.

**Weaknesses:**

1. While the paper claims to establish a new state-of-the-art, SpinQuant, as discussed in the related work, appears to far outperform the proposed method. It would be beneficial if the authors could initialize their model from SpinQuant and then apply their method to demonstrate its effectiveness.
2. Line 141 mentions that GPTQ can cannot be used to improve other transformation-based methods. However, GPTQ can significantly enhance the performance of rotation-based transformation like SpinQuant and QuaRot.
3. The rationale for using progressive freezing instead of a dynamic regularization term (i.e., AdaRound) for optimizing rounding values is not clearly articulated, nor is it convincingly demonstrated why this method surpasses AdaRound. The explanations in Lines 075 to 077 lack substantial evidence.
4. The paper does not compare its progressive rounding approach with existing rounding methods such as AdaRound, AdaQuant, and FlexRound, missing an opportunity to justify the need for progressive rounding.
5. Lines 230 to 231 discuss the potential negative impacts of optimizing the scale parameter $s$ in quantization. However, experimental evidence is needed to substantiate that learning $s$ is detrimental, as the current argument does not conclusively link optimization of $s$ to performance degradation.
6. The paper optimizes significantly more parameters than previous methods using limited data, raising concerns about potential overfitting. A user study or chat test could provide essential validation.
7. Line 473 states a significantly larger value than AWQ/OmniQuant, but it's unclear what this comparison signifies as these methods do not optimize rounding to flip variables.
8. The necessity of the sigmoid function $\sigma(\cdot)$ in Equation (9) should be experimentally validated. Why not use $\mathbf{v} \times \mathbf{s} \times (\theta^{q}-\mathbf{z})$ with $\mathbf{v}$ initialized to $\mathbf{1}$?
9. Further evaluation is needed on the effect of larger $t$ values in the expression $\frac{1}{\exp(tx)}$ and other scheduling functions in the **PAR Schedule** to demonstrate the method's robustness to different schedules.
10. More comparisons with block-wise reconstruction or learning-based methods, e.g, DuQuant (https://arxiv.org/abs/2406.01721) and QLLM (https://arxiv.org/pdf/2310.08041) could help confirm its effectiveness.
11. The author does not provide results for the proposed method without using transformation-based initialization.

**Questions:**

Why didn't the authors compare their method with OmniQuant for weight-activation quantization?

---

> ### Author Response · Authors · 2024-11-19
> **Reply to Reviewer wxS8**
>
> We sincerely thank you for providing such detailed feedback to improve our manuscript. The detailed response is attached below. Feel free to discuss it with us if you have any further questions that prevent you from accepting this work.
>
> > Q1: While the paper claims to establish a new state-of-the-art, SpinQuant, as discussed in the related work, appears to far outperform the proposed method. It would be beneficial if the authors could initialize their model from SpinQuant and then apply their method to demonstrate its effectiveness.
>
> Ans: Thanks for the suggestion. First, we want to clarify that SpinQuant and QuaRot both use the rotation transformation to enhance the weight-activation quantization by removing activation outliers. The function of our work is more similar to GPTQ, which is used to optimize weights that can be combined with SpinQuant. To show our method can be combined with SpinQuant, we conduct experiments on LLaMA-3-8B W4A4KV16 using SpinQuant official learned matrices [here](https://drive.google.com/drive/folders/1R2zix4qeXBjcmgnJN1rny93cguJ4rEE8) and export it to the code framework that we experimented with, (which we found has slightly higher wikitext perplexity than their paper reported in our code framework, presumably due to [SpinQuant](https://github.com/facebookresearch/SpinQuant/blob/main/eval_utils/main.py#L35) uses one fewer online Hadarmard transformation than [QuaRot](https://github.com/spcl/QuaRot/blob/main/fake_quant/main.py#L32-L45)). The results are shown below:
>
> | Method               | WT2      | C4        | Avg Acc   |
> |----------------------|----------|-----------|-----------|
> | SpinQuant            | 24.26    | 35.20     | 47.88     |
> | SpinQuant + GPTQ     | 8.68     | 14.37     | 60.91     |
> | SpinQuant + TesseraQ | **8.29** | **13.39** | **64.30** |
>
> > Q2: Line 141 mentions that GPTQ can not be used to improve other transformation-based methods. However, GPTQ can significantly enhance the performance of rotation-based transformation like SpinQuant and QuaRot.
>
> Ans: We apologize for any confusion here. In our original text, we meant scale transformation only rather than rotation transformation. Admittedly, the text may cause some confusion to readers, therefore, we revise Section 3.1 (line 141) to properly describe the problem statement. We change it to “*However, this
> Technique (GPTQ) makes it hard to improve scale-transformed models like AWQ*” and add a footnote saying that “*GPTQ can be combined with rotation-transformed models like QuaRot. We also compare it in Sec. 4.*”
>
>
> > Q3: The rationale for using progressive freezing instead of a dynamic regularization term (i.e., AdaRound) for optimizing rounding values is not clearly articulated, nor is it convincingly demonstrated why this method surpasses AdaRound. The explanations in Lines 075 to 077 lack substantial evidence.
>
> > Q4: The paper does not compare its progressive rounding approach with existing rounding methods such as AdaRound, AdaQuant, and FlexRound, missing an opportunity to justify the need for progressive rounding.
>
> Ans: Thank you for the question. We noticed that this concern was raised by other reviewers. Therefore we added a section in Appendix A to compare different rounding methods and illustrate why PAR is better than original rounding learning methods. We also refer you to our General Response to check our results for the rounding optimization ablation study. The results we demonstrate in General Response show that our rounding choice is better than the other two rounding methods both in layer-wise and block-wise reconstruction objectives. For instance, AdaRound with block-wise objective only obtains 9.05 perplexity on Wikitext2 while our method has 6.82 perplexity. In summary, our PAR consistently outperforms the other rounding methods regardless of objective. We think the reason is that we explicitly control the hardness of rounding variables through the progressive approach, which is better for handling the billion-level parameter space.
>
> As for FlexRound, we have checked the paper but we found they focus on traditional CNN PTQ mainly. The LLMs results are done on OPT models with 8-bit quantization, which is different from the scenarios where we comparing.  We also found out that FlexRound is not compatible with the code framework that we experimented with ([LLMC](https://github.com/ModelTC/llmc/tree/main)). Given the limited time in rebuttal, we are not able to implement that method on our own. However, we do find that SignRound (Cheng et al., 2023) demonstrates that SignRound is able to achieve higher performance than FlexRound. In light of this observation, we believe that our method can outperform FlexRound on LLMs.

---

> ### Author Response · Authors · 2024-11-19
> **Reply to Reviewer wxS8 (Part 2)**
>
> > Q5: Lines 230 to 231 discuss the potential negative impacts of optimizing the scale parameter in quantization. However, experimental evidence is needed to substantiate that learning is detrimental, as the current argument does not conclusively link optimization to performance degradation.
>
> Ans: Changing the scale parameter in the quantization step would fundamentally change the overall flooring results in Eq. (4), and thus will significantly affect the rounding variable optimization. As an intuitive example, assume the flooring result of $w_1/s$ is 2 and the rounding variable $\sigma(\nu_1)$ is optimized towards 1. In this case, by changing $s$, the flooring result could be 3 and thus fundamentally change the gradient direction of the rounding variable. As a result, the optimization process becomes unstable. This problem is stated in the original AdaRound (Nagel et al., 2020) work that scales should be fixed in the rounding optimization. To enable optimization, we adjust the dequantization scale factor, which is decoupled from the quantization step and does not require STE.
>
> > Q6: The paper optimizes significantly more parameters than previous methods using limited data, raising concerns about potential overfitting. A user study or chat test could provide essential validation.
>
> Ans: Thank you for your advice. To demonstrate the capability of our model, we follow OmniQuant (Xiao et al., 2023) to compare the instruction-tuned models for chatbots. The models are evaluated on the Vicuna benchmark. We use LLaMA-2-7B-chat and compress it by W3A16g128 weight quantization. Using GPT-4 evaluation, our TesseraQ model has a 59% win rate against the OmniQuant checkpoint.
>
> > Q7: Line 473 states a significantly larger value than AWQ/OmniQuant, but it's unclear what this comparison signifies as these methods do not optimize rounding to flip variables.
>
> Ans: Thanks for your question. This description validates our claim in the introduction section that to further improve existing scale-transformation and weight-clipping methods, adjustment to the weight tensor is necessary. The experiment here demonstrates that TesseraQ is able to optimize a huge amount of parameters in low-bit PTQ, which is why we improve over AWQ/OmniQuant. We meant to provide an intuition of what TesseraQ optimizes and also demonstrate the layer-wise difference in rounding optimization. Nevertheless, we agree that the original sentence may cause some confusion. We have revised the sentence to “*This proves that weight tensor-wise adjustment can be used to significantly improve previous scale transformation adjustments like AWQ.* ” to improve clarity.
>
> > Q8: The necessity of the sigmoid function in Equation (9) should be experimentally validated. Why not use $v\times s\times (\theta^q - z) $ with v initialized to 1.
>
> Ans: Thank you for this question, we chose this option because we try to avoid more hyper-parameter tuning of the learning rate of this scale parameter. If we directly optimize the scale factor, it could go to zero or even negative values if the learning rate is not well-tuned, making the learning process extremely unstable, a similar reason why AdaQuant does not perform well on LLMs.  Using sigmoid reparameterization, which is also applied to rounding variables, can apply the same learning rate for all variables. Besides, sigmoid reparameterization does not bring any cost in deployment, it can be safely merged into the original scale factor for the quantized checkpoint.
>
> > Q9: Further evaluation is needed on the effect of larger $t$ values in the expression $\frac{1}{exp(tx)}$ and other scheduling functions in the PAR Schedule to demonstrate the method's robustness to different schedules.
>
> Ans: Thanks for the advice, we additionally experimented with $t=6$ and $t=7$ and updated the results in Figure 2. To summarize, the$t=6$ and $t=7$ slightly degrade the performance from the optimal $t$, with 0.2 higher perplexity results and 0.2% lower average accuracy. We emphasize that this performance variation is much smaller when compared to the gap between AWQ and TesseraQ performance. In practice, we apply the same PAR schedule to all experiments and observe consistent results. Considering the small performance variation between different t and the limited time given in the rebuttal, we can only conduct more experiments regarding different PAR schedules in the next revision of our paper.

---

> ### Author Response · Authors · 2024-11-19
> **Reply to Reviewer wxS8 (Part 3)**
>
> > Q11: The author does not provide results for the proposed method without using transformation-based initialization.
>
> Ans: Thank you for the question. To clarify, transformation methods (like AWQ/OmniQuant) adjust weight distributions globally, while TesseraQ performs precise per-weight adjustments within a constrained range. These approaches are complementary - global adjustment first, then local refinement. Our goal is to demonstrate that a fine-grained adjustment after the global coarse adjustment can significantly improve performance. Without transformation-based initialization, our fine-grained weight adjustment approach may not optimize fully and yield sub-optimal performance. We would like to emphasize that TesseraQ is a booster on top of transformation-based initialization as mentioned in the contribution of the paper.
>
> > Q12: Why didn't the authors compare their method with OmniQuant for weight-activation quantization?
>
> Ans: Thank you for your question. Originally we did not put it since we only have LLaMA-3 model results and it is time-consuming to implement OmniQuant on them. As suggested by some reviewers, we realized it is not comprehensive enough to only include LLaMA-3 models. Therefore, in the newly updated Table 3, we include the OmniQuant performance on LLaMA-1/2 as well. As can be seen, previous approaches, such as AWQ, OmniQuant, QLLM, and OS+ all obtain similar results (49~51% average accuracy). In comparison, our method (initialized on AWQ) significantly improves the W4A4 performance by a huge margin (55% average accuracy).

---

> > ### Comment · Reviewer_wxS8 · 2024-11-19
> >
> > Thank the authors for the detailed response. However, I still have several concerns that need to be addressed:
> >
> > - SpinQuant primarily learns the R1 and R2 matrices and utilizes the standard Hadamard matrix as R4, which is not included in the provided link. Given that R4 can be derived as magic numbers (which I noted is accessible in the author's framework), I would appreciate it if the authors could include results using the complete version of SpinQuant. It is very important whether TesseraQ could further enhance SpinQuant's already impressive performance.
> >
> > - FlexRound has shown results on LLaMA; thus, I believe a comparison is essential.
> >
> > - Regarding SignRound, I am curious to know if the authors use the transformation process and learn the dequantize scale.
> >
> > - There seems to be a misunderstanding concerning my main question, which revolves around why progressive rounding is more effective in managing the billion-level parameter space. The discussion based solely on final performance is insufficient. Could the authors provide a detailed, possibly theoretical, analysis?
> >
> > - The scheduling functions for TesseraQ are incomplete, lacking non-monotone or monotonically increasing functions.
> >
> > - For Q5, the justification provided is unconvincing as optimizing the quantized scale simultaneously (e.g., LSQ) covers a larger space, potentially leading to better performance. The rationale used in AdaRound might not be appropriate here due to different optimization strategies.
> >
> > - Comparisons and initializations with DuQuant are still missing.
> >
> > - Could the authors provide some chat cases comparing OmniQuant vs. TesseraQ (with initialization from OmniQuant)?
> >
> > - I remain curious about the experimental results for Q8 and Q11.
> >
> > After these issues are addressed, I am prepared to increase the score. Given that there is more than a week left in the rebuttal phase, I believe this provides the authors sufficient time to prepare the necessary results.

---

> ### Author Response · Authors · 2024-11-20
> **Clarification on SpinQuant**
>
> We want to thank reviewer wxS8 for his/her quick reply. While we are actively running the additional experiments, we'd like to clarify the problem with SpinQuant first. The SpinQuant rotation setup is not the same as that of QuaRot. Indeed, R1 and R2 are learned, and R3/R4 are inserted as online Hadamard rotation. However, note that R4 is only inserted for *KV cache quantization*, which is both implemented here in [SpinQuant](https://github.com/facebookresearch/SpinQuant/blob/main/eval_utils/main.py#L150) and in [QuaRot](https://github.com/spcl/QuaRot/blob/main/fake_quant/main.py#L120). When applying W4A4KV16 quantization, which is the case in our rebuttal experiment, R4 is not inserted.
>
> The difference between QuaRot and SpinQuant lies in R3 online transformation, where QuaRot applies partial online Hadamard Transform to input of V_proj layers, and applies full Hadamard rotation matrix to O_proj layers weights [here](https://github.com/spcl/QuaRot/blob/main/fake_quant/rotation_utils.py#L226), while SpinQuant does not insert the online transformation to V_proj and rather applies rotation to O_proj weights and V_proj weights both in head dimension [here](https://github.com/facebookresearch/SpinQuant/blob/main/eval_utils/rotation_utils.py#L119).
>
> In our experiments, we directly save the FP16 checkpoint after the SpinQuant rotation matrix [here](https://github.com/facebookresearch/SpinQuant/blob/main/eval_utils/main.py#L31) to ensure we have the accurate FP16 checkpoint and then run experiments under our framework. We believe our results already use the full performance of SpinQuant.

---

> ### Comment · Reviewer_wxS8 · 2024-11-20
> **Reply for the Clarification on SpinQuant by Reviewer wxS8**
>
> I have checked the paper (v2 of SpinQuant, the latest version before iclr submission deadline). R4, inserted for ``down_proj`` is indeed employed for w4a4kv16 and R3 is only used for the kv cache quantization. **The names of the rotation matrix in the clarification given by the authors are different from the SpinQuant paper (v1, v2, and v3), which makes me feel so confused.** Therefore, I hope to see the results for SpinQuant with R1, R2, and R4 employed.
>
> By the way, the authors claim that their results already used SpinQuant's full performance, which is not convincing. Specifically, R1+R2+R4 achieves 7.1 wikitext PPL for w4a4kv16, which I think is the ideal performance. **Additionally, through only employing R1 and R2, the PPL achieved by the authors is about 6 points higher than the paper (v3).** I ran SpinQuant's original code two months ago and can get a similar performance in its paper.

---

> > ### Author Response · Authors · 2024-11-23
> > **Reply to Reviewer wxS8 (Round 2)**
> >
> > Thanks for your comment. Please check our detailed reply below. Note that to differentiate the questions in round 1 and round 2, we use “Q1*” to denote the question in the second round.
> >
> > > Q1*: Therefore, I hope to see the results for SpinQuant with R1, R2, and R4 employed. Additionally, through only employing R1 and R2, the PPL achieved by the authors is about 6 points higher than the paper (v3).
> >
> > Thank you for your reply. We truly apologize for confusing R3 and R4 and your previous question. Let's use the v3 version of SpinQuant notation. To clarify, our previous experiments were already conducted with W4A4KV16 quantization with R1, R2, R4 enabled since SpinQuant's GitHub only supports this setting. There is no option to turn off the online transformation (i.e., R4).
> >
> > After carefully debugging, we have located why our SpinQuant does not match the paper performance. We applied per-channel asymmetric weight quantization while SpinQuant repo uses per-channel symmetric weight quantization. After changing it to symmetric weight quantization, the performance gap is largely minimized. A slight difference still exists between our LLMC framework and SpinQuant repo, due to (1) some numerical protection code in handling zero-point and scale, (2) SpinQuant uses per-head activation quantization [here](https://github.com/facebookresearch/SpinQuant/blob/main/eval_utils/main.py#L120) while our method adopts per-token quantization for all layers.
> > Our new results are shown below. Note that the results we conducted for SpinQuant, SpinQuant+GPTQ, and SpinQuant + TesseraQ used the LLMC framework to ensure a fair comparison. The results in parentheses are quoted from the paper (v3).
> >
> > | Method               | WT2        | C4        | Avg Acc   |
> > |----------------------|------------|-----------|-----------|
> > | SpinQuant            | 8.45 (7.7) | 14.17     | 61.10     |
> > | SpinQuant + GPTQ     | 7.60 (7.1) | 12.61     | 63.96     |
> > | SpinQuant + TesseraQ | **7.25**   | **11.89** | **64.45** |
> >
> > > Q2*: FlexRound has shown results on LLaMA; thus, I believe a comparison is essential.
> >
> > Ans: Thanks for the suggestion. FlexRound has results on W4A16 quantization performance. To compare it, we apply TesseraQ on LLaMA-7B with W4A16 quantization (initialized from AWQ). Additionally, we quote the results of AdaRound in the FlexRound paper. The accuracies on five commonsense reasoning tasks and wikitext2 perplexity are shown below.
> >
> > | Method                              | Wikitext2 | Avg Acc.  |
> > |-------------------------------------|-----------|-----------|
> > | 7B Pretrain                         | 5.67      | 64.53     |
> > | 7B AWQ+TesseraQ                     | **5.80**  | **64.04** |
> > | 7B FlexRound                        | 9.18      | 61.32     |
> > | 7B AdaRound (Quoted from FlexRound) | 9.69      | 61.84     |
> >
> > > Q3*: Regarding SignRound, I am curious to know if the authors use the transformation process and learn the dequantize scale.
> >
> > Ans: SignRound is essentially AdaQuant with the SignSGD method, in which they use STE gradient estimation and apply signSGD to update the parameters with only direction information. In SignRound, the quantization step size is learned while no transformation is incorporated. In our reply to Q9* below, we have a performance of no-transformation results and we still have a better performance than SignRound.

---

> > > ### Author Response · Authors · 2024-11-23
> > > **Reply to Reviewer wxS8 (Round 2) (Part 2)**
> > >
> > > > Q4*: There seems to be a misunderstanding concerning my main question, which revolves around why progressive rounding is more effective in managing the billion-level parameter space. The discussion based solely on final performance is insufficient. Could the authors provide a detailed, possibly theoretical, analysis?
> > >
> > > Ans: Finding optimal rounding parameters for neural network quantization is known to be NP-Hard, as it requires solving a binary integer programming problem over billions of variables. Branch-and-bound (B&B) has emerged as one of the most effective algorithmic frameworks for solving such discrete optimization problems by systematically exploring and pruning the solution space. Our PAR method inherits key theoretical properties from B&B. Like B&B, PAR systematically reduces the solution space through a hierarchical decision process. At each iteration, PAR employs a scoring mechanism HS(ν) = |σ(ν) - 0.5| analogous to B&B's bounding function - both guide the selection of promising variables and ensure the solution converges toward optimality. This hierarchical optimization allows PAR to inherit B&B's ability to break down complex combinatorial problems into manageable subproblems while maintaining solution quality.
> > >
> > > However, PAR differs crucially from B&B in its design for billion-parameter LLMs. While B&B explores multiple branches by trying both 0 and 1 values and requires exponential memory to maintain the search tree, PAR adopts a confidence-based, one-way branching strategy. It makes irreversible decisions guided by the scoring metric and allows remaining soft variables to compensate through continuous optimization. This adaptation preserves B&B's systematic reduction of the solution space while making the optimization tractable for LLM-scale problems.
> > >
> > > We have tried our best to come up with a theoretically-driven justification for the reviewer’s question using the B&B analogy. We believe a full theoretical analysis for PAR is out of scope for now, and we will investigate it in our future work.
> > >
> > > > Q5*: The scheduling functions for TesseraQ are incomplete, lacking non-monotone or monotonically increasing functions.
> > >
> > > Ans: To address this concern, we experimented with a monotonically increasing function $1-\frac{1}{exp((1–x)t)}$. We run $t=2$ and $t=3$, the soft rate progression becomes
> > >
> > > `Soft_Rate = [13.5, 15.0, 16.6, 18.5, 20.5, 22.8, 25.3, 28.1, 31.2, 34.6, 38.4, 42.6, 47.3, 52.5, 58.2, 64.6, 71.7, 79.6, 88.3, 98.0]` for t=2, `Soft_Rate = [5.0, 5.8, 6.8, 8.0, 9.3, 10.9, 12.7, 14.9, 17.4, 20.3, 23.8, 27.8, 32.5, 38.0, 44.4, 51.9, 60.7, 70.9, 83.0, 97.0]` for t=3.
> > >
> > > We summarize the results of these two schedules on LLaMA-2-7B W2A16g128 quantization in the table below. We can find that monotonically increasing scheduling functions will degrade the performance of PAR with more than 1 average perplexity increase and a 2% average accuracy decrease (our handcrafted scheduling has 8.8 average perplexity and 58.4 average accuracy). In Section 3.2 we have emphasized that progressively slowing down the soft rate will yield better performance.
> > >
> > > | Scheduling t                     | Avg PPL | Avg Acc.  |
> > > |----------------------------------|---------|-----------|
> > > | $t = 2$ (monotonically increasing) | 9.82    | 56.72     |
> > > | $t = 3$ (monotonically increasing) | 10.42   | 54.67     |
> > >
> > > > Q6*: For Q5, the justification provided is unconvincing as optimizing the quantized scale simultaneously (e.g., LSQ) covers a larger space, potentially leading to better performance. The rationale used in AdaRound might not be appropriate here due to different optimization strategies.
> > >
> > > Ans: Our optimization strategy is different from AdaRound. However, the optimization space is similar, i.e., optimizing the sigmoid reparameterization of rounding variables. To validate this, we experimented with this LSQ method which uses STE to train scale parameters together with rounding variables optimization of TesseraQ. We observed consistent NAN loss in the 25~30th block on LLaMA-2-7B W2A16g128 quantization. We also found that the loss value is much higher than using fixed scale in TesseraQ quantization.
> > >
> > > | Method                               | Optimize $s$ | Fix $s$ |
> > > |--------------------------------------|--------------|---------|
> > > | Block-Output Loss Value @ 25th block | 1080         | 606     |
> > >
> > >
> > > > Q7*: Comparisons and initializations with DuQuant are still missing.
> > >
> > > Ans:  Thanks for your suggestion, we run the DuQuant experiments on LLaMA-3-8B W4A4 quantization. We implement our rounding learning on top of the DuQuant framework. Since DuQuant is also a block-reconstruction algorithm, our rounding learning happens right after DuQuant within each block. The results are summarized in the following table. Note that the accuracy is averaged from 6 tasks (including BoolQ following DuQuant paper evaluation).

---

> > > > ### Author Response · Authors · 2024-11-23
> > > > **Reply to Reviewer wxS8 (Round 2) (Part 3)**
> > > >
> > > > | Method             | Wikitext2 | Avg Acc from 6 tasks.  |
> > > > |--------------------|-----------|------------------------|
> > > > | DuQuant            | 8.56      | 66.21                  |
> > > > | DuQuant + LWC      | 8.06      | 67.72                  |
> > > > | DuQuant + TesseraQ | **7.69**  | **69.10**              |
> > > >
> > > > > Q8*: Could the authors provide some chat cases comparing OmniQuant vs. TesseraQ (with initialization from OmniQuant)?
> > > >
> > > > Ans: Thanks, we upload some chat cases in Appendix B.4 of our paper, please check it.
> > > >
> > > > > Q9*: I remain curious about the experimental results for Q8 and Q11.
> > > >
> > > > Ans: Thanks for the suggestion. We run these two experiments on LLaMA-2-7B W2A16g127 again.
> > > > For Q8, we run the dequantization scale tuning without sigmoid reparameterization. Without sigmoid reparameterization, the learning process becomes much more sensitive to the learning rate. Therefore, we experimented with 3 learning rates: {1e-5, 3e-6, 1e-6}. The results are shown below. For LR=1e-5, the block reconstruction loss becomes NAN in the 30th block. While for the other two lower learning rates, the performance is lower to the sigmoid reparameterization version (6.82 wikitext2 perplexity and 58.37 average accuracy.)
> > > >
> > > >
> > > > | LR   | WikiText2               | C4                      | Avg Acc                 |
> > > > |------|-------------------------|-------------------------|-------------------------|
> > > > | 1e-5 | Stopped due to NAN loss | Stopped due to NAN loss | Stopped due to NAN loss |
> > > > | 3e-6 | 7.23                    | 11.96                   | 56.35                   |
> > > > | 1e-6 | 7.25                    | 11.81                   | 57.75                   |
> > > >
> > > > For Q11, we run our method without AWQ transformation. The performance is quite close to one with AWQ transformation. Here, we also compare against SignRound and observe better performance.
> > > >
> > > >
> > > > | Method                                       | WikiText2 | C4    | Avg Acc |
> > > > |----------------------------------------------|-----------|-------|---------|
> > > > | AWQ initialization + TesseraQ                | 6.82      | 10.77 | 58.35   |
> > > > | TesseraQ only                                | 6.90      | 11.16 | 57.98   |
> > > > | SignRound (No transformation initialization) | NaN       | NaN   | 55.92   |

---

> > > > > ### Comment · Reviewer_wxS8 · 2024-11-24
> > > > >
> > > > > Thanks again for the reply. However, some of my concerns are still left:
> > > > > * SpinQuant learns the rotation matrix under its quantization settings. However, the authors perform their method on the model from SpinQuant with different quantization settings (numerical protection code in handling zero-point and scale; per-head activation quantization in SpinQuant). I think this is very easy to implement with a few lines of code. Therefore, I think the results are not based on a fair comparison (also different settings to test will be harmful to the performance).
> > > > > * I agree that giving a full theoretical analysis in this phase is difficult. However, I still want to ask how this method helps ensure the rounding values converge closely to 0 or 1 in the soft phases.
> > > > > * I also find that the experimental settings in the paper and here may be unfair. For example, SignRound should employ transformation tricks, however, the authors compare this to their methods with transformation initialization. If the authors want to emphasize their method as a plug-and-play tool, I think SignRound can also be that tool and should be compared in this way.
> > > > > * I am also curious about whether the activation clipping is employed for QuaRot in this paper, which is also essential to get the complete version of that method.
> > > > >
> > > > > Due to the above reasons, I will still think about it for a while to make a decision.

---

> ### Author Response · Authors · 2024-11-26
> **Reply to Reviewer wxS8 (Round 3)**
>
> We appreciate reviewer wxS8’s detailed comment on TesseraQ. Before we answer your questions, we would like to strengthen several points.
>
> 1. The reviewer kept claiming we made an unfair comparison of related works like SpinQuant. However, any mismatch in SpinQuant+GPTQ also exists on our SpinQuant+TesseraQ. **We did not take any advantage of setting mismatch in SpinQuant.** The same logic applies to QuaRot with activation clipping. The experimental settings of QuaRot, QuaRot+GPTQ, and QuaRot+TesseraQ are the same. **Thus the universality of experimental settings across different methods makes our comparison and results fair**.
>
> 2. Based on the reviewer's question, it seems like the reviewer remains curious about the performance of TesseraQ under all prior transformation-based methods available online. However, we have tested in many cases (AWQ, OmniQuant, QuaRot, SpinQuant, No initialization). In all cases, we have demonstrated TesseraQ’s ability to consistently improve the existing method’s quantization. Compared to other weight quantization work, we have done a more comprehensive analysis. We believe our experiments in the original paper and post-rebuttal in the revised version are sufficient and demonstrate TesseraQ’s performance-boosting capabilities on prior methods.
>
> 3. It seems like the reviewer focuses on several cases but ignores our other major experiments. In Table 1/2, we have demonstrated a significant improvement in 2-bit weight-only quantization. Yet the reviewer maintains a clear rejection opinion **because we lack the complete version of weight-activation transformation results in Table 3, which is orthogonal to our major contribution that focuses on weight-only quantization.** More importantly, we have established TesseraQ’s superior performance compared to prior works like QuaRot, OmniQuant, and AWQ in the pre-rebuttal submission. In post-rebuttal, the reviewer requested to add SpinQuant results. We would like to note that SpinQuant is an arXiv paper, not yet published, and the reviewer’s criticism about our paper’s results being unfair based on an arXiv work is not appropriate. Having said that, we have recreated experiments from SpinQuant with our framework and have provided sufficient justification and qualitative/quantitative comparison. Across all experiments, we have shown the TesseraQ’s effectiveness.

---

> > ### Author Response · Authors · 2024-11-26
> > **Reply to Reviewer wxS8 (Round 3) (Part 2)**
> >
> > > Q: SpinQuant learns the rotation matrix under its quantization settings. However, the authors perform their method on the model from SpinQuant with different quantization settings (numerical protection code in handling zero-point and scale; per-head activation quantization in SpinQuant). I think this is very easy to implement with a few lines of code. Therefore, I think the results are not based on a fair comparison (also different settings to test will be harmful to the performance).
> >
> > Ans: Thank you for your suggestion. We understand the reviewer’s comment and would like to emphasize that aligning the setting across different frameworks took us a lot of time because it requires tons of debugging and manual checking of the original SpinQuant framework. We still maintain that our results in the previous rebuttal round (Q1*) is fair. Here, after carefully aligning the settings in SpinQuant by matching the numerical protection code and per-head activation quantization, we re-evaluate the performance below.
> >
> > | Method               | WT2  |
> > |----------------------|------|
> > | SpinQuant            | 7.71 |
> > | SpinQuant + GPTQ     | 7.09 |
> > | SpinQuant + TesseraQ | 6.98 |
> >
> > >Q: I agree that giving a full theoretical analysis in this phase is difficult. However, I still want to ask how this method helps ensure the rounding values converge closely to 0 or 1 in the soft phases.
> >
> >
> > Ans: Thanks for clarifying the question. The reviewer is correct that the softening phase does not ensure the close convergence to 0 or 1 in rounding variables. In fact, this convergence is ensured by the harden phase. In every iteration of PAR, we harden some variables to 0/1, and the remaining soft variables are optimized, this allows the reconstruction loss to gradually decrease. At the end, the harden phase ensures all variables are rounded to 0/1.
> >
> > In the last iteration of PAR, the soft variables remaining are extremely low (e.g., 0.5% in our schedule). In this case, rounding them to 0/1 does not affect the block reconstruction loss. Here is an example of showing the block reconstruction loss when calibrating the LLaMA-2-7B with W2A16g128 quantization.
> > We measure the hard rounding loss before PAR and at the last iteration of PAR before and after the hardening. As we can see, with iterative soften and harden phases, the reconstruction loss is largely reduced (from col 1 to col 2), and in the last iteration, the hardening of the remaining 0.5% soft variables does not impact the reconstruction loss.
> >
> > | Block Idx | Loss before PAR begins | Loss without hardening the last 0.5% soft variables | Loss after hardening the last 0.5% soft variables |
> > |-----------|------------------------|-----------------------------------------------------|---------------------------------------------------|
> > | 1         | 0.1397763              | 0.02278                                             | 0.02275                                           |
> > | 2         | 144.29                 | 0.4867                                              | 0.7680                                            |
> > | 3         | 2.0709                 | 1.0669                                              | 1.1003                                            |
> > | 4         | 3.7673                 | 2.2060                                              | 2.2213                                            |
> > | 5         | 6.5995                 | 3.9875                                              | 3.9946                                            |
> >
> >
> > >Q: I also find that the experimental settings in the paper and here may be unfair. For example, SignRound should employ transformation tricks, however, the authors compare this to their methods with transformation initialization. If the authors want to emphasize their method as a plug-and-play tool, I think SignRound can also be that tool and should be compared in this way.
> >
> > A: We run SignRound experiments on the LLaMA-2-7B W2A16g128 setting using AWQ initialization. Check the results below.
> >
> > | Method          | WikiText2 | Avg Acc |
> > |-----------------|:---------:|:-------:|
> > | SignRound + AWQ |    8.70   |  56.80  |
> > | TesseraQ + AWQ  |    6.82   |  58.37  |
> >
> > We would like to note that the reviewer’s feeling about our experiments being unfair is not right. In the previous round (Q9*), the experiments conducted comparing TesseraQ and SignRound both without transformation initialization already showed our superior performance. The above table compares them both with AWQ transformation initialization again showing the superiority of TesseraQ. All our experiments conducted so far pre- and post-rebuttal have claimed that TesseraQ is a plug-and-play tool and can boost prior transformation methods. We hope the reviewer acknowledges our method’s effectiveness given the comprehensive experiments provided.

---

> > > ### Author Response · Authors · 2024-11-26
> > > **Reply to Reviewer wxS8 (Round 3) (Part 3)**
> > >
> > > > Q: I am also curious about whether the activation clipping is employed for QuaRot in this paper, which is also essential to get the complete version of that method.
> > >
> > > A: We also implemented the 0.9 clipping ratio version here on LLaMA-2-7B. We will switch all our QuaRot experiments to the 0.9 clipping ratio in our next revision of the paper in Table 3. Also, we want to emphasize the fact that we include the QuaRot + GPTQ on weight-only quantization in Table 1, where TesseraQ still outperforms QuaRot + GPTQ by a large margin. This weight-only quantization result should not be ignored. In fact, all Table 1/2 results show TesseraQ’s superior performance with comprehensive prior work comparison.
> > >
> > > | Method            | Bitwidth |  WT2  |   C4  | Avg Acc |
> > > |-------------------|:--------:|:-----:|:-----:|:-------:|
> > > | QuaRot            |   W4A4   | 13.01 | 18.65 |  48.62  |
> > > | QuaRot + GPTQ     |   W4A4   |  6.00 |  8.21 |  61.66  |
> > > | QuaRot + TesseraQ |   W4A4   |  5.95 |  8.01 |  61.87  |
> > > | QuaRot            |   W3A3   |  6551 |  5944 |  35.12  |
> > > | QuaRot + GPTQ     |   W3A3   | 11.62 | 18.70 |  47.15  |
> > > | QuaRot + TesseraQ |   W3A3   | 10.06 | 13.71 |  52.40  |

---

> ### Comment · Reviewer_wxS8 · 2024-11-27
>
> Thanks again for the concise reply. I do not have any further questions. The authors have clearly made a substantial effort to conduct thorough analyses and experiments to refine their method. Considering the large volume of LLM quantization papers published this year, I find that the most notable contributions are those that provide clear, practical enhancements in terms of performance or speed. **Many papers claim state-of-the-art results; therefore, it is crucial to compare this method against their full versions to ascertain its true value. Although a plug-and-play method might improve existing models, its effectiveness could be limited if the full versions already deliver strong results. Furthermore, the results reported in the paper show significant divergence from previous studies.** For instance, 1) the similar Perplexity (PPL) performance of SpinQuant+GPTQ versus SpinQuant+TesseraQ does not demonstrate improvements, as discussed in numerous papers. Additional testing on more datasets and models (similar to the full version of QuaRot) is necessary to fully evaluate its impact. 2) The authors have chosen not to use activation clipping for QuaRot-RTN, and the results including activation clipping also diverge significantly from those reported in its paper. The PPL for LLaMa-2-7B is 13.01 reported by the authors in the rebuttal phase, which is approximately **5 points higher** than the figure in Table 3 of QuaRot's latest version, where QuaRot-RTN even incorporates KV cache quantization. A similar discrepancy is observed with downstream tasks, e.g., about **6 points lower** on PIQA.
>
> **Considering that there may be other results in the paper that exhibit the same issues mentioned above (I have no time to check one-by-one), and given that the rebuttal does not allow for major revisions, I do not recommend the paper for acceptance at ICLR at this time. I suggest addressing the aforementioned problems, particularly aligning the results with previous studies and replacing the results of baselines with those of their full version, which I believe are serious issues.** Additionally, providing a detailed description of the experimental setup would help clarify the methodology and strengthen the paper's contribution.
>
> Note: Apologies for requesting a comparison with SpinQuant (submitted to arXiv in May), but I could not find the specific policy regarding this matter on the ICLR website. It is worth mentioning that this helps me observe the above concerns.

---

> > ### Comment · Reviewer_wxS8 · 2024-11-27
> > **Follow-up (Round 4)**
> >
> > As a reviewer, I acknowledge the authors' contribution to the algorithm, but I believe there are significant flaws in the experimental setup.

---

### Official Review · Reviewer_EUJb · 2024-10-23

**Soundness:** 2
**Presentation:** 2
**Contribution:** 2
**Rating:** 5
**Confidence:** 4

**Summary:**

Large language models (LLMs) has recently gained widespread attention，but face challenges in memory and computation requirements.  Post-training quantization (PTQ) is a common approach to address these issues. This work mainly proposes some optimization methods for PTQ targeting LLM.The proposed TesseraQ, a novel PTQ technique, can quantize LLMs to ultra-low bits. It introduces progressive adaptive rounding to iteratively transition soft rounding variables to hard ones during reconstruction and optimizes the dequantization scale parameters. TesseraQ can be integrated with existing PTQ algorithms like AWQ and OmniQuant. Experiments show that TesseraQ significantly improves performance.

**Strengths:**

The paper proposes a new rounding optimization design called TesseraQ， it can be combined with other PTQ methods to achieve new performance levels, establishing a new technological level for quantifying LLM in terms of confusion, downstream accuracy, and hardware performance.

**Weaknesses:**

1.  3.2 Some expressions are not very clear, and some variables have not been explained
2.  The new rounding error optimization method proposed in the paper mentions that compared to previous methods ( https://arxiv.org/pdf/2004.10568 ) does not rely on regularization loss , but HS and sorting were introduced. It doesn't have a significant advantage.For example, computational complexity and convergence speed.
3.  The paper combines the proposed rounding error optimization method with other methods for experimentation and comparison with other quantization methods, but does not use other rounding error optimization methods for the same experiment, and does not demonstrate the superiority of this method.For example, the Adaround mentioned in the article.
4. When comparing quantitative indicators such as W4A4/W3A3 that include activation, the QuaRot initial method was used, but it was not compared with using the QuaRot method alone. The advantage of the proposed new rounding error optimization method in improving accuracy cannot be clearly demonstrated.Suggest to use QuaRot as a baseline separately to include the results.

**Questions:**

1. The new rounding error optimization method proposed in the paper mentions that compared to previous methods ( does not rely on regularization loss , but HS and sorting were introduced. Does it have advantages in training convergence speed and video memory usage?
2. The paper combines the proposed rounding error optimization method with other methods for experimentation and comparison with other quantization methods, but does not use other rounding error optimization methods for the same experiment, Can relevant experiments be supplemented ?
3.When comparing quantitative indicators such as W4A4/W3A3 that include activation, the QuaRot initial method was used, but it was not compared with using the QuaRot method,Can relevant experiments be supplemented?

---

> ### Author Response · Authors · 2024-11-19
> **Reply to Reviewer EUjb**
>
> We thank the reviewer for the constructive comments and suggestions. We hope our response will address your concerns, see detailed comments below.
>
> > Q1: 3.2 Some expressions are not very clear, and some variables have not been explained.
>
> Ans: We apologize for any unclear expressions or unexplained variables. We would like to explain the core idea of Progressive Adaptive Rounding (PAR) here. The main idea of PAR is to optimize the binary rounding variables effectively. To tackle this problem, we relax them into continuous variables by applying the sigmoid function, so that these continuous variables can be updated using block reconstruction loss and gradient descent. However, we expect all variables to be binary after optimization, therefore, we propose a progressive approach, that hardens partial variables at a time and then optimizes the remaining *soften* variables for loss optimization. The learned rounding variables are 0/1, which can be merged into original weights, and then we can maintain the original quantization function for deployment.
>
> We are happy to improve any expressions that the reviewer finds unclear.
>
> > Q2: The new rounding error optimization method proposed in the paper mentions that compared to previous methods (https://arxiv.org/pdf/2004.10568) does not rely on regularization loss, but HS and sorting were introduced. It doesn't have a significant advantage. For example, computational complexity and convergence speed.
>
> Ans: Thank you for the question. In terms of computational complexity, our method is indeed similar to the original AdaRound. As for training convergence, we argue that our PAR is better when applied to LLMs. In Appendix A, we add a new set of ablation studies on different rounding methods, including the original AdaRound and the STE-based optimization, AdaQuant. Our results demonstrate that PAR is the best choice in LLM PTQ.
>
> > Q3: The paper combines the proposed rounding error optimization method with other methods for experimentation and comparison with other quantization methods, but does not use other rounding error optimization methods for the same experiment, and does not demonstrate the superiority of this method. For example, the Adaround mentioned in the article.
>
> Ans: Thanks for the question. Again we refer you to our General Response and Appendix A for a fair comparison where we initialize all experiments with the same AWQ checkpoint. The results we demonstrate in General Response show that our rounding choice is better than the other two rounding methods both in layer-wise and block-wise reconstruction objectives. For instance, AdaRound with block-wise objective only obtains 9.05 perplexity on Wikitext2 while our method has 6.82 perplexity. In summary, our PAR consistently outperforms the other rounding methods regardless of objective. We think the reason is that we explicitly control the hardness of rounding variables through the progressive approach, which is better for handling the billion-level parameter space.
>
>
> > Q4: When comparing quantitative indicators such as W4A4/W3A3 that include activation, the QuaRot initial method was used, but it was not compared with using the QuaRot method alone. The advantage of the proposed new rounding error optimization method in improving accuracy cannot be clearly demonstrated. Suggest to use QuaRot as a baseline separately to include the results.
>
> Ans: Thank you for your suggestions. In Table 3 we already provided the QuaRot baseline results. To summarize, we show that TesseraQ + QuaRot is better than GPTQ + QuaRot, which is again better than QuaRot itself. Moreover, we update Table 3 to include LLaMA-1/2-7B models as well to show a more comprehensive comparison. For example, the QuaRot results for LLaMA-3-8B are here:
>
> | Method            | Bitwidth | Wikitext2 ($\downarrow$) | Avg Acc ($\uparrow$) |
> |-------------------|----------|--------------------------|----------------------|
> | QuaRot            | W4A4     | 17.83                    | 51.83                |
> | QuaRot + GPTQ     | W4A4     | 8.39                     | 62.87                |
> | QuaRot + TesseraQ | W4A4     | 8.05                     | 65.12                |
> | QuaRot            | W3A3     | 9.1e4                    | 35.25                |
> | QuaRot + GPTQ     | W3A3     | 93.08                    | 37.87                |
> | QuaRot + TesseraQ | W3A3     | 27.80                    | 47.33                |

---

> ### Comment · Reviewer_EUJb · 2024-11-26
>
> thank for your reply， I keep my score

---

> > ### Author Response · Authors · 2024-11-27
> > **Reply to Reviewer EUJb**
> >
> > Dear Reviewers, we thank you for your time and efforts in reviewing our manuscript and reading each other’s discussion.
> > We noticed that you retain negative opinions after reading our rebuttal. We are happy to provide a revision if there is any further concern regarding our work. If the reviewer agrees with Reviewer wxS8, we'd like to emphasize that we have made a new round of responses to his/her concerns regarding the unfair or unclear experimental results. As you will observe, all our results show TesseraQ’s superior capability with fair comparison and comprehensive analysis. We look forward to you considering raising your scores if our response addresses your queries.

---

### Official Review · Reviewer_cbEs · 2024-10-30

**Soundness:** 3
**Presentation:** 4
**Contribution:** 3
**Rating:** 6
**Confidence:** 3

**Summary:**

The paper presents a novel method to compress large language models (LLMs) into ultra-low-bit formats while maintaining high performance. It introduces Progressive Adaptive Rounding (PAR), which gradually decides rounding schedules for groups of parameters within LLMs while retaining the training flexibility via Sigmoid-based soft-hard transition, improving quantization stability. By integrating with block-wise reconstruction instead of layer-wise optimization, TesseraQ captures inter-layer dependencies more effectively, and de-quantization scale tuning further reduces reconstruction errors. The method integrates seamlessly with existing quantization approaches like AWQ and OmniQuant, significantly improving perplexity and accuracy across benchmarks. TesseraQ achieves state-of-the-art results, offering a practical solution for deploying LLMs efficiently in memory-constrained environments.

**Strengths:**

The paper offers an interesting and effective add-on for block-wise reconstruction for LLM quantization with optimized rounding policy. Especially for Ultra-low bit scenarios, such combination advances the such field where prior quantization methods struggle to maintain performance for various tasks.

The methodology is well-designed, with clear theoretical motivation for each component, including PAR and dequantization scale tuning. The experimental results are robust, demonstrating TesseraQ’s superiority across multiple benchmarks such as WikiText2 and downstream reasoning tasks. The paper carefully evaluates TesseraQ’s effectiveness by comparing it to several existing methods, providing quantitative evidence that supports the proposed contributions.

In terms of the overall writing quality, the paper is generally well-organized and explains technical concepts like rounding optimization and block reconstruction in a clear and structured manner.

The significance of the paper lies in its potential impact on LLM deployment, particularly for memory-constrained environments such as consumer devices and edge processors. TesseraQ pushes the boundaries of ultra-low-bit quantization, a growing need as LLMs become larger and more ubiquitous.

**Weaknesses:**

1. The paper proposes PAR instead of the common STE implementation. There is a lack of justification why direct utilization of STE for the rounding policy estimation would result in worse performance so that PAR is significant in the overall design.

2. The paper purposes an effective rounding optimization add-on for enhancing the block-wise quantization for LLM, but there is a lack of theoretical analysis of the different between: A) BLOCK-WISE & Normal Rounding (round to the nearest) and  B) the proposed BLOCK-WISE & Rounding Optimization. If A) is well trained and fully converged, would it theoretically have the similar mathematical foundations as B)?

**Questions:**

The paper is generally clear and well-written, the algorithm has been fully presented. While according to my perspective in the weakness section, I have the following question:

1. Why does the paper adopt Progressive Adaptive Rounding (PAR) instead of the more common Straight-Through Estimator (STE) for rounding policy estimation? What specific limitations or performance issues with STE justify the need for PAR in the overall design? If STE is applied, whether such progressive-stage training is still required?

2. Could the authors justify the theoretical basis for the improvement achieved by rounding optimization over the baseline?

---

> ### Author Response · Authors · 2024-11-19
> **Reply to Reviewer cbEs**
>
> Thank you for your positive feedback on our work, we will try our best to address your questions, please check our detailed reply below.
>
> > Q1: The paper proposes PAR instead of the common STE implementation. There is a lack of justification why direct utilization of STE for the rounding policy estimation would result in worse performance so PAR is significant in the overall design.
>
> Ans: Thank you for the suggestion. We noticed that this is a general question raised by multiple reviewers so we conduct experiments in Appendix A, please check our general response. Regarding the common STE implementation, we assume the reviewer is asking for AdaQuant (Hubara et al., 2021). We agree that STE, in theory, has higher potential since they do not limit the optimization space to binary choices. However, the problem with AdaQuant is its unstable tuning. AdaQuant directly finetunes the floating-point weights, which is trained via discrete quantization function in forward calculation and STE in backward calculation. Consequently, AdaQuant faces instant change in forward and estimated gradient in backward,  which becomes extremely sensitive to the learning rate selection and may require different learning rates for different layers/blocks in the model.
>
> In AdaRound original paper (Nagel et al., 2020) Table 5, the authors compare their method with STE and find out AdaRound outperforms the STE approach. Similarly, in our reproduced experiments in Appendix A, we found that AdaQuant obtains extremely unstable results.
>
> > Q2: The paper purposes an effective rounding optimization add-on for enhancing the block-wise quantization for LLM, but there is a lack of theoretical analysis of the different between: A) BLOCK-WISE & Normal Rounding (round to the nearest) and B) the proposed BLOCK-WISE & Rounding Optimization. If A) is well-trained and fully converged, would it theoretically have similar mathematical foundations as B)?
>
> Ans: Thanks for your question. We’d like to clarify that option (A) Block-wise & Normal Rounding can be regarded as the same methodology in OmniQuant (Shao et al., 2023). They calibrate the transformation scales as well as the weight clipping range using the BLOCK-WISE objective while maintaining the normal rounding function. As shown in our experiments, especially in Table 1 first several rows, TesseraQ optimized on OmniQuant checkpoint (option B) clearly improves the original OmniQuant performance (option A) in W2A16 quantization. This indicates that B) has higher potential than A). For mathematical foundation explanations, we also refer you to our reply to your Q4, which uses an simple example to demonstrate effectiveness of rounding optimization.
>
> > Q3: Why does the paper adopt Progressive Adaptive Rounding (PAR) instead of the more common Straight-Through Estimator (STE) for rounding policy estimation? What specific limitations or performance issues with STE justify the need for PAR in the overall design? If STE is applied, whether such progressive-stage training is still required?
>
> Ans: Please check our answer to Q1. In short, STE-based calibration is less stable and requires careful hyperparameter tuning. On the other hand, our progressive approach always optimizes continuous soft variables and explicitly controls the overall hardness. Therefore, our method is easier to optimize in practice.
>
> > Q4: Could the authors justify the theoretical basis for the improvement achieved by rounding optimization over the baseline?
>
> Ans: Thank you for your question. Here we use an example from AdaRound (Nagel et al., 2020) to theoretically demonstrate why rounding-to-nearest (RTN) is not optimal. The RTN approach essentially minimizes the weight space distance while we need to consider the loss function space minimization. In PTQ, this can be realized by applying Taylor expansion to the loss function so that our actual objective is $\Delta W^\top H^W \Delta W$, where $\Delta W$ is the distance between quantized weight and FP weight and $H^W$ is the Hessian matrix with respect to weights. Now consider a two-variable example $W=[w_1, w_2]$ and the Hessian matrix $H^W = [[1, 0.5], [0.5, 1]]$. In such a case, the overall objective becomes $\Delta w_1^2 + \Delta w_2^2 + \Delta w_1\Delta w_2$. We can find that the term $\Delta w_1 \Delta w_2$ requires a different sign of $\Delta w_1$ and $\Delta w_2$ to minimize the overall objective. In this case, rounding to the nearest may not give optimal results if $\Delta w_1 \Delta w_2$ is positive. Therefore, finding the optimal rounding choice to minimize the objective is necessary. For more details, we refer the reviewer to the original AdaRound paper.

---

> ### Comment · Reviewer_cbEs · 2024-11-25
>
> Thanks for the authors' careful and detailed responses.
>
> I have tried the purposed PAR instead of direct STE in one of my previous project where also a series of binary 0 or 1 is expected from the model output. (If it was implemented correctly) I could see a little slowdown of the convergence time of my model (probably due to the progressive design) but the the output series is a more stable than my previous STE implementation where some batches won't not result in a relatively-huge changes in the output behavior.
>
> I think the most of my concerns are taken care of by the authors. I also have gone through carefully all the comments and reviews, thus I decide to maintain my score.
>
> Thank you for your effort again.

---

> > ### Author Response · Authors · 2024-11-27
> > **Reply to Reviewer cbEs**
> >
> > Thank you so much for engaging with us. We appreciate your feedback and believe your suggestions have greatly improved our paper. We look forward to you considering raising your scores if our response addresses your queries.

---

### Official Review · Reviewer_co5y · 2024-11-04

**Soundness:** 3
**Presentation:** 3
**Contribution:** 2
**Rating:** 6
**Confidence:** 5

**Summary:**

This paper introduces TesseraQ, a post-training quantization method for LLMs that focuses on optimizing weight rounding parameters using block reconstruction techniques. The key design is a Progressive Adaptive Rounding approach that iteratively converts soft rounding variables to hard variables, along with dequantization scale tuning.
﻿
The method is designed to work with existing quantization approaches like AWQ and OmniQuant, significantly improving their performance particularly for ultra-low bit scenarios (2-4 bits). Results show major improvements in both perplexity metrics and downstream task accuracy across different LLaMA models and quantization schemes.
﻿
The main contribution is enabling better ultra-low bit quantization of LLMs through tuning of quantization rounding while maintaining model performance, demonstrated through comprehensive empirical results.

**Strengths:**

1. The paper presents a clear motivation for ultra-low bit LLM quantization, with a well-justified technical solution. The authors effectively demonstrate how current methods struggle with low-bit quantization and explain why optimizing the entire weight tensor is necessary. The progressive adaptive rounding approach is sound to handle the problem.

2. The writing quality is exceptionally clear and professional. Complex technical concepts are explained systematically, from basic quantization principles to advanced optimization techniques. The figures and algorithms are informative and well-designed, particularly Figure 1 which clearly illustrates the workflow.

3. The experimental results show substantial improvements over strong baselines. For example, improving WikiText2 perplexity from 37.37 to 8.05 on LLaMA-2-7B with W2A16 quantization is a significant achievement. The ablation studies effectively isolate the contributions of PAR and DST.

4. The practical utility is demonstrated through real-world evaluations. The authors test different hardware configurations, measure actual throughput, and analyze memory consumption.

**Weaknesses:**

1. The paper lacks detailed technical comparisons with similar methods. While mentioning prior approaches like AdaRound and AdaQuant,  it doesn't experimentally demonstrate why the proposed methods are superior to the these methods. The rationale behind several design choices (sigmoid reparameterization; harden score metric in Eq 6; avoiding scale optimization in the quantization step) isn't fully justified through comparative experiments.

2. The evaluation demonstration has problems.
- The inconsistency in model selection between Tables 1, 2 and 3 makes it difficult to draw comprehensive conclusions.
- Runtime and GPU memory analysis is limited to the 7B model.  As the runtime and memory consumption are important metircs, the authour should provide comprehensive efficiency metrics across different model sizes.
- The paper would be stronger with complete results for newer models like LLaMA-3 across all experiments.

3. The technical novelty is relatively incremental. Both block reconstruction and scale tuning are well-established in PTQ literature. While the paper makes useful improvements, it builds primarily on existing techniques rather than introducing fundamentally new concepts.

**Questions:**

1. The choice to restrict rounding to ±1 steps seems limiting. Why not allow multiple step sizes as in AdaQuant? This decision could impact optimization flexibility, especially for ultra-low bit quantization where precision is crucial. Has this design choice been empirically validated against alternatives?

2. The effectiveness with weight transformation techniques is unclear. Specifically, would TesseraQ maintain its benefits when applied after Hadamard matrix transformations (as in QuaRot)?

---

> ### Author Response · Authors · 2024-11-19
> **Reply to Reviewer co5y**
>
> Thank you for acknowledging the clarity and significance of our work. We appreciate your comments and would like to address your concerns below.
>
> > Q1: The paper lacks detailed technical comparisons with similar methods. While mentioning prior approaches like AdaRound and AdaQuant, it doesn't experimentally demonstrate why the proposed methods are superior to these methods. The rationale behind several design choices (sigmoid reparameterization; harden score metric in Eq 6; avoiding scale optimization in the quantization step) isn't fully justified through comparative experiments.
>
> A1: We understand your concern regarding some detailed design choices of our work. First, regarding the comparison to AdaRound/AdaQuant, we add an ablation study of rounding optimization choices in Appendix A. Please check our general response and the updated manuscript for more details.
>
> Second, other design choices have mostly been addressed in the existing literature. We explain the insights here. Regarding the sigmoid reparameterization, this is the standard relaxation step in {0, 1} binary space which is essential to differentiate the rounding variables and provide a smooth loss landscape [1]. For the reason of avoiding scale optimization in the quantization step, we did this since changing the scale would fundamentally change the overall flooring results in Eq. (4), and thus will result in a significant change in rounding variables gradients. This problem is stated in the original AdaRound (Nagel et al., 2020) work that scales should be fixed in the rounding optimization. To enable optimization, we adjust the dequantization scale factor, which is not affected by quantization operation and does not require STE.
>
> Last, we agree with the reviewer that the Harden score (HS) metric could be interesting to study. We conduct experiments to compare the random selection of Harden variables and the selection based on the HS score. The results are shown below
>
> | Harden methods→              | Random | HS score sorting |
> |------------------------------|--------|------------------|
> | LLaMA-2-7B W2g128 Perplexity | 7.62   | 6.82             |
>
> > Q2: The inconsistency in model selection between Tables 1, 2 and 3 makes it difficult to draw comprehensive conclusions.
> Q3: The paper would be stronger with complete results for newer models like LLaMA-3 across all experiments.
>
> Ans: Thank you for your advice. We have updated Table 1 to include LLaMA-3 models and more baseline results like SignRound, and GPTQ with QuaRot. Meanwhile, for weight-activation quantization comparison, we updated Table 3 to add LLaMA-1/2-7B so that all LLaMA series are evaluated. It is worth noting that our method still outperformed other methods on these newly tested models. For example, in Table 3, we reported that our method achieves 55% average accuracy of LLaMA-2-7B W4A4 quantization, while all existing scale-transformation methods like QLLMs, AWQ, OS+, OmniQuant obtain 49~51% average accuracy.
>
> > Q3: Runtime and GPU memory analysis is limited to the 7B model. As the runtime and memory consumption are important metrics, the authors should provide comprehensive efficiency metrics across different model sizes.
>
> Ans: Thank you for this suggestion. Here we provide a detailed Runtime comparison and GPU memory on LLaMA-2-7B/13B/70B models when applying TesseraQ. Overall the GPU resources, and algorithm runtime scales with the calibration dataset size. For the 70B model, we can apply CPU offloading to block input/output data to save extra memory, at a little bit cost of extra runtime.
>
> | Calib Data Size | Batch Size | 7B GPU Mem/Runtime | 13B GPU Mem/Runtime | 70B GPU Mem/Runtime |
> |-----------------|------------|--------------------|---------------------|---------------------|
> | 128             | 1          | 17.5GB / 3.2h      | 23.9GB / 6.0h       | 54.0GB / 27.0h      |
> | 256             | 2          | 28.6GB / 3.9h      | 32.5GB / 7.3h       | 66.9GB / 34.3h      |
> | 512             | 2          | 40.4GB / 4.0h      | 53.8GB / 7.4h       | 74.5GB / 36.0h      |
> | 512             | 4          | 65.4GB / 6.0h      | 75.7GB / 11.6h      | 76.4GB*/ 47.1h      |
>
> *denotes apply CPU offloading to block input/output data

---

> ### Author Response · Authors · 2024-11-19
> **Reply to Reviewer co5y (Part 2)**
>
> > Q4: The technical novelty is relatively incremental. Both block reconstruction and scale tuning are well-established in PTQ literature. While the paper makes useful improvements, it builds primarily on existing techniques rather than introducing fundamentally new concepts.
>
> Ans: We would like to clarify the contribution of our work. Although the block-reconstruction and rounding optimization framework is proposed in the literature, few works have successfully implemented them on LLMs due to the challenging nature of the billion-level parameter space. Our proposed framework demonstrates that rounding optimization, if properly implemented with our progressive approach, can be used to significantly improve the LLM quantization performance. In the revision, we also add experiments to show the effectiveness of PAR over traditional weight optimization methods like AdaRound and AdaQuant. We have also revised Section 3.1 to emphasize this point in our problem statement.
>
> > Q5: The choice to restrict rounding to ±1 steps seems limiting. Why not allow multiple step sizes as in AdaQuant? This decision could impact optimization flexibility, especially for ultra-low bit quantization where precision is crucial. Has this design choice been empirically validated against alternatives?
>
> Ans: Thank you for your question. AdaQuant, in theory, has higher potential since they do not limit the optimization space to binary choices. However, the problem with AdaQuant is its unstable tuning.  AdaQuant directly finetunes the floating-point weights, which is trained via discrete quantization function in forward calculation and STE in backward calculation. Consequently, AdaQuant faces instant change in forward and estimated gradient in backward, which becomes extremely sensitive to the learning rate selection and may require different learning rates for different layers/blocks in the model.
>
> In AdaRound original paper (Nagel et al., 2020) Table 5, the authors compare their method with STE and find out AdaRound outperforms the STE approach. Similarly, in our reproduced experiments in Appendix A and General Response, we found that AdaQuant obtains extremely unstable results. Consequently, we use a lower learning rate but obtain less optimized results.
>
> > Q6: The effectiveness with weight transformation techniques is unclear. Specifically, would TesseraQ maintain its benefits when applied after Hadamard matrix transformations (as in QuaRot)?
>
> Ans: Yes, TesseraQ follows the standard uniform weight quantization rules after merging the learned rounding variables. When combined with QuaRot, we learn the rounding choices for the Hadamard transformed weight $\mathbf{H}\mathbf{W}$ and maintain the original benefits. After TesseraQ+QuaRot, we can use the same original QuaRot inference pipeline, i.e., uniform weight-activation quantization plus online FP32 hadamard transformation.
>
> [1] Maddison, Chris J., Andriy Mnih, and Yee Whye Teh. "The concrete distribution: A continuous relaxation of discrete random variables." ICLR 2017.

---

> > ### Comment · Reviewer_co5y · 2024-12-01
> >
> > Thanks for your reply. Cosidering the comments from other reviewers. I keep my score.

---

### Official Review · Reviewer_Ck8i · 2024-11-04

**Soundness:** 3
**Presentation:** 2
**Contribution:** 3
**Rating:** 5
**Confidence:** 3

**Summary:**

This paper presents a PTQ method，aTesseraQ, for LLM.  The author propose optimize weight rounding through a progressive
approach based on block reconstruction. Besides, the author also propose dequantization scale tuning to mitigate the reconstruction error.
The experiments on LLaMa models shows good performance, and this training-based PTQ method could be implemented with other PTQ methods.

**Strengths:**

1. The idea of optimizing the weight rounding error in LLM is interesting, while most previous methods focus on the weight clipping or distribution transmation.
2. The author proposes Progressive Adaptive Rounding strategy by deviding all rounding variables into harden and soften.
3. The performance in W2A16 setting is good and TesseraQ is orthogonal to other post-training quantization (PTQ) methods for LLM.

**Weaknesses:**

1. unclear writing: The writing in Section 3.2 of the paper is not sufficiently clear, making it somewhat difficult to read and understand. Moreover, this section constitutes the core content of the methodology and pertains to the detailed description of the method itself.
2. Outdated Comparison: The comparison methods in Table 1 appear to be somewhat outdated. The authors should compare with more advanced weight-only methods to demonstrate its superiority.
3. Besides, other tables comparison seems inadequate. The authos should also report the comparison results with other Rounding method, such as Adaround and signRound.
4. The author should discuss the method's limitation.

**Questions:**

1. Why the author didn't compare with Adaround and signRound in Table1 and other experiments?
2. Compared with the traditional Layer-wise Rounding method Adaround, what is the superiority of the proposed TesserQ, is the block-wise reconstruction granulity or the Harden and Soften design? Why? The author should add more related discussion and results to analysis it.

---

> ### Author Response · Authors · 2024-11-19
> **Reply to Reviewer Ck8i**
>
> Thank you for reviewing our work and providing useful suggestions. Please check our detailed reply to your questions/comments.
>
> > Q1: The writing in Section 3.2 of the paper is not sufficiently clear, making it somewhat difficult to read and understand.
>
> Ans: We apologize for any confused expression or equation in Section 3.2. We would like to explain the core idea of Progressive Adaptive Rounding (PAR) here. The main idea of PAR is to optimize the binary rounding variables effectively. To tackle this problem, we relax them into continuous variables by applying the sigmoid function, so that these continuous variables can be updated using block reconstruction loss and gradient descent. However, we expect all variables to be binary after optimization, therefore, we propose a progressive approach, that hardens partial variables at a time and then optimizes the remaining *soften* variables for loss optimization. The learned rounding variables are 0/1, which can be merged into original weights, and then we can maintain the original quantization function for deployment. We also modified the methodology section to improve its clarity. Feel free to suggest detailed modifications that we can make.
>
> > Q2: Outdated Comparison: The comparison methods in Table 1 appear to be somewhat outdated. The authors should compare with more advanced weight-only methods to demonstrate its superiority.
>
> Ans: Thanks for providing the feedback on Table 1. We did not report some baseline results since they missed some results of either LLaMA-1 or LLaMA-3. We are happy to provide a more detailed comparison with some most recent works in Table 1. We now add LLaMA-3-8B/70B quantization results and 2 additional baseline results, the SignRound and the GPTQ with QuaRot. Overall we find our method still outperformed additional baselines by a large margin, especially on smaller models and lower bitwidth. For example, on LLaMA-3-8B W2A16g128 quantization, our method obtains 10.03 perplexity while AWQ is 334 and the GPTQ with QuaRot is 17.4.
>
> > Q3: Besides, other tables comparison seems inadequate. The authos should also report the comparison results with other Rounding method, such as Adaround and signRound.
>
> Ans: Thanks for your suggestion. We agree that a rounding method comparison would be beneficial. We provided a new section in Appendix A to compare other rounding methods. We also explain our revision, please check our General Response. SignRound is now compared in Table 1 as well. For example, on LLaMA-2-13B W2A16g128 quantization, SignRound has 7.68 perplexity while our TesseraQ has 5.92 perplexity.
>
> > Q4: The author should discuss the method's limitation.
>
> Ans: One potential negative effect of TesseraQ would be its relatively higher calibration time compared to other layer-wise optimization methods. In Section 4.3, we report the calibration data size’s impact on accuracy and algorithm runtime. The LLaMA-2-7B model takes 3~6 hours depending on the data to finish the rounding optimization, which is larger than AWQ/GPTQ which is around 0.5 hours. However, we emphasize that TesseraQ is still faster than end-to-end quantization-aware training which requires significantly more GPU resources to compute and store gradients. We will leave accelerating the rounding optimization as our future development goal and discuss it in our conclusion section.
>
> > Q5: Why the author didn't compare with Adaround and signRound in Table1 and other experiments?
>
> Ans: We apologize for missing SignRound results in Table 1, which has been added for comparison right now. Notably our method is significantly better than SignRound in perplexity metric. For example, in LLaMA-2-7B W2g128 quantization scheme, our method obtains 6.82 perplexity while the SignRound method crashed and got NAN value.
>
> For AdaRound, it is designed originally for convolutional-based networks. As pointed out in GPTQ (Frantar et al., 2022), traditional rounding optimization methods are complex and challenging to scale to billions of parameters. In fact, there is no official AdaRound implementation for LLM, and SignRound (Cheng et al., 2023) provided their own implementation results in AdaRound, where they demonstrate that SignRound is able to achieve higher performance. In light of this observation, we believe that our method is able to outperform both rounding optimization methods. In addition, our new ablation study in Appendix A also confirms this observation. Please check our General response and the new Appendix A for more details.

---

> ### Author Response · Authors · 2024-11-19
> **Reply to Reviewer Ck8i (Part 2)**
>
> > Q6: Compared with the traditional Layer-wise Rounding method Adaround, what is the superiority of the proposed TesserQ, is the block-wise reconstruction granulity or the Harden and Soften design? Why? The author should add more related discussion and results to analysis it.
>
> Ans: Thank you for the suggestion. To answer your question, both block-wise reconstruction granularity and PAR algorithm choice contribute to the final performance. The results we demonstrate in General Response show that our rounding choice is better than the other two rounding methods both in layer-wise and block-wise reconstruction objectives. Additionally, block-wise objective works better than layer-wise objective as it considers the attention and MLP inter-layer activity. Nevertheless, even with the block-wise objective, the rounding optimization method is essential to improve the performance. For instance, AdaRound with block-wise objective only obtains 9.05 perplexity on Wikitext2 while our method has 6.82 perplexity. In summary, our PAR consistently outperforms the other rounding methods regardless of objective. We think the reason is that we explicitly control the hardness of rounding variables through the progressive approach.

---

> ### Comment · Reviewer_Ck8i · 2024-11-25
>
> Thank you very much for the authors' response, which addressed  most of my concerns. I have carefully read the comments from other reviewers, particularly the discussion from reviewer wxS8. I share similar concern regarding the supplementary compared experiments of the SpinQuant method, mainly focusing on the misaligned accuracy comparisons, which leaves me with some doubts. I acknowledge the contribution of this paper to LLM quantization from the perspective of rounding; however, due to the incompleteness of some experimental results, I maintain my original score.

---

> > ### Author Response · Authors · 2024-11-27
> > **Reply to Reviewer Ck8i**
> >
> > Dear reviewer, we thank you for your time and efforts in reviewing our manuscript and reading each other’s discussion. We encourage you to discuss any further concerns of our paper.
> >
> > Specifically, we noticed that you remained negative opinions due to the discussion we had with Reviewer wxS8. We want to emphasize that we have made a new round of responses to his/her concerns regarding the unfair or unclear experimental results. As you will observe, all our results show TesseraQ’s superior capability with fair comparison and comprehensive analysis. We look forward to you considering raising your scores if our response addresses your queries.

---

### Author Response · Authors · 2024-11-19
**General Response to All Reviewers**

We sincerely thank all reviewers for their thoughtful and constructive feedback. We are encouraged that they agree that our method is interesting, effective, and significantly improves the uniform quantization performance of LLMs. We will try our best to address each reviewer’s questions and concerns point-to-point. We also welcome any further discussion of our paper.

Here we would like to highlight some major changes to our paper:

1. **Updated more perplexity results to Table 1**. Following suggestions by Reviewer co5y, we run our TesseraQ on LLaMA-3-8B and 70B models and compare it against existing methods like GPTQ, AWQ. Moreover, as suggested by Reviewer Ck8i, co5y, we compare two additional baseline methods, SignRound, and GPTQ combined with QuaRot. We show that our method consistently outperforms the existing methods on all series of LLaMA models. For example, on LLaMA-3-8B W2A16g128 quantization, our method obtains **10.03** perplexity while the AWQ model is 334.1 and GPTQ+QuaRot is 17.43.
Some representative results are listed here:
| Method         | LLaMA-3-8B W2A16g128 | LLaMA-3-8B W3A16g128 | LLaMA-3-70B W2A16g128 | LLaMA-3-70B W3A16g128 |
|----------------|----------------------|----------------------|-----------------------|-----------------------|
| GPTQ + QuaRot  | 17.43                | 7.42                 | 30.89                 | 4.98                  |
| AWQ            | 334.1                | 8.24                 | 10.98                 | 4.63                  |
| TesseraQ + AWQ | **10.03**            | **6.90**             | **7.47**              | **4.13**              |

2. **Updated more WA quantization results to Table 3**. Following suggestions by Reviewer co5y and wxS8, we additionally run TesseraQ on LLaMA-1/2-7B models for weight-activation quantization for comprehensive comparison. In addition, we add two more baseline results like Atom and QLLM in Table 3. The results demonstrate that TesseraQ can be used to initialize both scale-transformation and rotation-transformation models in weight-activation quantization scenarios, achieving even better performance. For example, on W4A4 quantized LLaMA-2-7B, our method obtains **55.12%** average accuracy while previous methods like AWQ, OmniQuant, OS+, and QLLM obtain 49~51% average accuracy.

3. **Provided an ablation study on rounding optimization**. As suggested by all reviewers, a comparison to existing rounding learning methods like AdaRound, and AdaQuant, is needed. To clarify, very few works have tried implementing rounding optimization on LLMs quantization and GPTQ (Frantar et al.) claimed that these methods *are complex and challenging to scale to billions of parameters*. Nevertheless, we agree that a more practical study would benefit our work. Therefore, we implemented AdaRound and several variants to demonstrate the effectiveness of our proposed method with LLaMA-2-7B W2A16g128 quantization. Please check our new results in Appendix A. We also summarize the results in the following table. In summary, our PAR consistently outperforms the other two rounding methods regardless of objective. We think the reason is that we explicitly control the hardness of rounding variables through the progressive approach. On the other hand, AdaRound and AdaQuant are less optimized on LLMs and may require more hyper-parameter search.
| Rounding          | Objective | Wikitext2 ($\downarrow$) | C4 ($\downarrow$) |
|-------------------|-----------|--------------------------|-------------------|
| None (AWQ)        | Layer     | 14.65                    | 18.67             |
| AdaRound          | Layer     | 10.68                    | 15.67             |
| AdaQuant          | Layer     | 16.78                    | 21.34             |
| PAR               | Layer     | **9.43**                 | **12.79**         |
| None (OmniQuant)  | Block     | 11.06                    | 16.34             |
| AdaRound          | Block     | 9.05                     | 11.45             |
| AdaQuant          | Block     | 10.05                    | 14.87             |
| PAR               | Block     | **6.82**                 | **10.77**         |

4. **Revise Sections 3.1 and 3.2 for better clarity**. We justify the reason why we chose the rounding optimization framework and the reason we introduced Progressive Adaptive Rounding.

For other technical questions, please check our detailed response to each reviewer.

---

### Meta-Review · Area_Chair_3vXJ · 2024-12-20

**Metareview:**

The paper introduces TesseraQ, a novel post-training quantization (PTQ) technique for large language models (LLMs), aiming to improve performance by optimizing weight rounding through a block reconstruction approach. The authors also propose a progressive adaptive rounding strategy and the optimization of dequantization scale parameters. Experimental results show that TesseraQ significantly improves perplexity and downstream task accuracy when applied to the LLaMA-2 model, particularly with ultra-low bit quantization. The authors also highlight that TesseraQ can be seamlessly integrated with existing PTQ methods, offering a flexible and effective solution for optimizing LLMs.

The strengths of this paper lie in its innovative approach to weight rounding and the proposed progressive adaptive rounding strategy. The experiments are well-conducted, showing that TesseraQ outperforms previous methods, such as AWQ, on multiple quantization schemes. Additionally, the ability to integrate TesseraQ with existing PTQ methods enhances its practical applicability. The results, particularly in terms of perplexity reduction and downstream task performance, are promising and contribute to advancing the field of LLM optimization.

However, several concerns were raised by reviewers that the authors attempted to address but did not fully resolve. One major issue is the inconsistency in the reported results, especially in comparison to similar methods like SpinQuant and QuaRot. The reported perplexity for LLaMA-2-7B is notably higher than previous benchmarks, which raises doubts about the generalizability of the method. The authors have not sufficiently aligned their results with these previous studies, and the lack of detailed comparison with full versions of baseline methods undermines the claimed improvements. Additionally, the experimental setup is not sufficiently detailed, leaving unclear how some of the discrepancies arose.

Given these unresolved issues, particularly the divergence in results and the absence of a thorough comparison with full baseline versions, I recommend rejecting the paper at this time. While the proposed method is promising and the experimental results are noteworthy, the issues with result alignment, insufficient comparisons, and lack of methodological clarity need to be addressed before this work can be considered for acceptance. A more thorough evaluation across different datasets and models, along with an in-depth experimental setup description, would strengthen the paper significantly.

**Additional Comments On Reviewer Discussion:**

During the rebuttal period, reviewers raised concerns about the reported results, specifically the discrepancies in perplexity and downstream task performance compared to prior methods such as SpinQuant and QuaRot. A key issue was the significant divergence in results, particularly in terms of perplexity for the LLaMA-2-7B model and downstream task accuracy. Additionally, reviewers requested that the authors align their results with those from the full versions of baseline methods, as the reported improvements did not demonstrate clear advantages over existing approaches.

The authors addressed these concerns by clarifying their experimental setup and providing additional explanations for the observed discrepancies. They emphasized the differences in experimental conditions, such as the use of activation clipping in QuaRot-RTN, which was not applied in their experiments. Despite these clarifications, reviewers remained unconvinced by the responses, as the differences in results were still notable and could not be fully reconciled. The authors also did not sufficiently provide comparisons with the full versions of the baseline methods, as requested.

In my final decision, I weighed the authors' efforts to clarify their methodology and experimental setup, but the remaining discrepancies in results and the lack of a thorough comparison with baseline full versions led to a recommendation for rejection. The paper introduces an interesting method, but the unresolved issues regarding result alignment and comparative analysis significantly hindered its acceptance.

---

### Decision · Program_Chairs · 2025-01-22

Reject